# Bat dynamics modelling as a tool for conservation management in subterranean environments

Dragoş Ştefan Măntoiu[1]*, Ionuț Cornel Mirea[2,3], Ionuț Cosmin Şandric[4], Alina Georgiana Cîşlariu[5], Iulian Gherghel[6,7,8], Silviu Constantin[2,3], Oana Teodora Moldovan[1,3]

1 "Emil Racoviţă" Institute of Speleology—Cluj Department, Cluj-Napoca, Romania, 2 Department of Geospeleology and Paleontology, "Emil Racoviţă" Institute of Speleology, Bucharest, Romania, 3 Romanian Institute of Science and Technology, Cluj-Napoca, Romania, 4 Faculty of Geography, Department of Environmental Studies, University of Bucharest, Bucharest, Romania, 5 Faculty of Biology, Department of Botany and Microbiology, University of Bucharest, Bucharest, Romania, 6 Department of Exact Sciences and Natural Sciences, Institute of Interdisciplinary Research, "Alexandru Ioan Cuza" University of Iaşi, Iaşi, Romania, 7 Faculty of Natural and Agricultural Sciences, Ovidius University Constanţa, Constanţa, Romania, 8 Department of Biology, Case Western Reserve University, Cleveland, OH, United States of America

* stephen.mantoiu@gmail.com

**Data Availability Statement:** All relevant data are within the paper and its Supporting Information files.

## Abstract

Bat species inhabit subterranean environments (e.g., caves and mines) in small areas with specific microclimatic conditions, during various periods of their life cycle. Bats can be negatively influenced by microclimatic changes within their roosts if optimal habitat patches become unavailable. Therefore, proper management solutions must be applied for the conservation of vulnerable bat populations, especially in show caves. We have pursued an ensemble species distribution modelling approach in subterranean environments to identify sensible patches for bats. Using multi-annual temperature monitoring and bat distribution surveys performed within ten caves and mines, including show caves, we modelled relevant habitat patches for five bat species. The temperature-based variables generated from this approach proved to be effective when processed via species distribution models, which generated optimal validation results, even for bats that were heavily clustered in colonies. Management measures are proposed for each show cave to help in long-time conservation of hibernation and maternity colonies. These measures include creating suitable microclimatic patches within the caves by ecological reconstruction measures, tourist management practices in relation to bats, and show cave fitting recommendations. This approach has never been performed at this scale due to the complex geostatistical challenges involving subterranean environment mapping and can be further used as best practice guidelines for future conservation projects.

## Introduction

Subterranean environments (SE: caves, mines) are one of the most widespread habitats worldwide and harbour diverse ecosystems with a high degree of species endemism [1]. Although

**Funding:** The field data was obtained during the grants CAVEMONITOR (EEA Grant 17SEE/2014) and 146/2010 "Mapping subterranean sites and evaluating the conservation status of bat species from the most important sites in the Buila-Vânturariţa National Park". A grant of the Ministry of Research, Innovation and Digitization, CNCS/CCCDI – UEFISCDI, project no. 2/2019 (DARKFOOD), within PNCDI III and the EEA Financial Mechanism 2014-2021, under the project contract no. 3/2019 (KARSTHIVES) financially supported the different teams for the analyses and results interpretation. A scholarship from the Romanian Academy within the doctoral school program (SCOSAAR) was awarded to DSM, and helped in data collection within the initial stages of the project. IG was partially funded by the Romanian Ministry of Research, Innovation and Digitization, CNCS – UEFISCDI, through project number: PN-III-P1-1.1-PD-2021- 0591, within PNCDI III.

**Competing interests:** The authors have declared that no competing interests exist.

many SEs benefit from some form of conservation status, revolving around water resources or emblematic species, there is no global uniform classification regarding their conservation priorities [2] with some examples of systematic conservation classifications within North America [3].

Bats are strongly linked to SEs, as more than half of these species use them as roosts in various periods of their life cycles [4], in turn providing crucial ecosystem services in almost all biomes [5]. The conservation priorities of SEs include maintaining stable bat populations as key components of their ecosystems, with mentions in the EUROBATS agreement [6], the United States Endangered Species Act 1973 [7], and the RELCOM 2007 (Latin American Bat Research Network). EUROBATS has classified some caves or mines within Europe as important underground sites for bats based on species richness schemes and population size, yet in some cases, the status has not been translated to local laws, especially for artificial underground sites.

Specific regional indicators show promising results regarding the population trends of cave-dwelling bat species [8], but general global trends have recorded a decline throughout their ranges, according to the IUCN Red List of Threatened Species [9] or focal species related studies [10], with examples of irreversible systematic cave disruptions in biodiversity hotspots [11]. Therefore, greater interest was offered to some cave-dwelling bat species in legislation and conservation efforts, following the recommendations of international guidelines [12].

Cave-dwelling bats can be subject to anthropogenic pressures, especially in show caves, as large populations usually concentrate in small habitat patches. These populations offer valuable ecosystem services for natural habitats, but also for agriculture and human health purposes, controlling the populations of invertebrates that can pose significant issues for both the natural and economic environments [13]. Altering small habitats such as SEs can have a significant effects on the integrity of the local and regional ecosystems, therefore, specific conservation measures need to be applied to ensure the long-time survival of cave-dwelling bats [14].

Cave microclimatic conditions strongly influence bat populations [15–17] as they require optimal habitat patches within their roosts to ensure a successful hibernation or nursing cycle. The animals balance their energy reserves during the cold season [14, 18], and search for hotter areas during the nursing season [19]. The stable temperature values recorded in deep sectors of SEs often reflect the exterior annual average temperatures [20, 21]. Variations in deep sectors can still occur where seasonal and daily airflows shift between multiple entrances or underground watercourses flow [22]. During the cold season, heterothermic temperate insectivorous bats reduce their metabolic rate to survive prolonged periods of food scarcity [23, 24]. This is achieved by altering various torpor bout durations with brief arousal episodes depending on temperature variations in the roost and the exterior [23, 25–27]. Some bat species, such as *Rhinolophus* spp., strongly prefer SEs to support their key yearly biological requirements [28]. They are not strictly bound to these habitats, as their ecological plasticity allows for torpor in various roost types with similar environmental conditions, but most of their populations prefer SEs and therefore, are referred to as cave-dwellers [28]. Furthermore, roost fidelity for these populations is high, especially during key periods such as hibernation or maternity [29]. This is additionally influenced by the SE topography, network type, relative air humidity, vapour condensation, atmospheric composition, and the amount of drip water that penetrates the system [30]. High microclimatic variability within a roost can reduce the size of suitable torpor patches for some cave-dwelling bat species during the cold season, while favouring the crevice-dwelling bat species [31]. Hibernation areas are selected in specific sectors of SEs at various heights, depending on bat body weight, fat reserves, fitness, but also with regards to inter and intra specific interactions, forming conspecific or heterospecific aggregations [32]. Bats show various species-specific individual clustering abundances to conserve energy [24].

During a hibernation season, they can exit torpor and move several times within the roost or between a cluster of roosts in search of favourable climatic conditions [33]. The animals can increase their arousal episodes and even forage near the roost if the air temperature reaches a certain threshold (e.g., approximately 10˚C for *R. ferrumequinum* [27]).

Anthropogenic factors can cause disturbances for hibernating or nursing bats, often expressed as mass tourism. These interventions can lead to strong disturbances in the energy equilibrium of torpid bats, such as weight loss [34], which can lower the survival rate of the exposed population [35] or the abandonment of optimal nursing sites [36]. Open space cave-dwelling bats are especially vulnerable to external factors which can cause arousals, such as artificial lighting, changes in air circulation, and slight increases in air temperature, which mainly occur during visiting hours [37, 38]. On a global scale, climatic changes may also influence air circulation in caves, leading to increased air temperatures and loss of suitable habitats for a broad group of cave-dwelling species [39]. The recent climatic evolution correlated with anthropogenic factors amplifies the need for extensive spatial biodiversity and microclimatic monitoring within SEs, as a framework for wildlife and habitat conservation [40].

Desipite the vast distribution of SEs worldwide and their large species diversity [1], spatial distribution studies regarding subteranean habitats remain scarce. Species distribution models (SDM) can help predict potentially suitable habitats within a site and can be used as conservation and management tools [41], but SDMs have rarely been applied to SEs due to a series of challenges in building environmental variables [42]. The existing studies presented limited results for the spatial microclimatic preferences of animals in SEs, often for a single cave or for a short time frame [43, 44]. Some approaches focused on collecting occurrences from within the sites and projecting them on external environmental variables [41, 45, 46], allowing for an increased number of analysed sites at the expense of detailed habitat mapping for each particular SE. Cave-dwelling bats use specific areas within a SE for shelter or reproduction [47] and are conditioned by a limited set of environmental factors, mostly revolving around stable microclimates.

Although bats are one of the most commonly studied taxons in SEs, especially within the temperate regions [48], conservation efforts seem to have mixed results, pointing out that more systematic site specific studies are required, or that more complex disitrbution recognition patterns of their roosting preferences need to be undertaken to implement successful conservation measures.

Given these limitations, our goal was to map the distribution of several cave-dwelling bat species during multiple key periods of their life cycle and to understand how the animals use the sites to propose optimal management solutions, especially in show caves. All bat species within our studied range are strictly protected and most of the selected roosts are included within the EUROBATS important underground sites list. The species are included in the European Natura 2000 Network, and therefore require specific conservation measures to ensure population connectivity and habitat integrity, including the creation of new protected natural areas [49].

We used species distribution models (SDM), specifically ensemble SDMs (ESDM) to reduce biases of individual modelling techniques by averaging predictions across a multitude of models [50]. It is generally accepted that this approach provides a better predictive performance compared to a single model running algorithm [51–53]. We created a series of temperature-based variables extracted from a systematic air temperature monitoring effort. We further compared the results to the observed anthropogenic impact for each site to identify relevant conservation and management measures and help reduce the impact on bat populations. We hypothesized that bats' spatial and temporal distribution within SEs can be accurately predicted via SDMs, aiding future management efforts. This novel approach was challenged by a

series of limiting factors associated with creating subterranean environmental spatial datasets and reliable occurrence sampling strategies. Results can be used as a conservation tool for bat colonies that inhabit caves and mines in most bioregion, especially show caves, aiding managers in their conservation efforts.

## Methods

This study was performed in compliance with the recommendations described within the Bat Surveys Good Practice Guidelines of the Bat Conservation Trust. The field protocol was authorised by the Romanian Academy—Natural Monuments Commission (#3660/22.11.2012). The data collection efforts were designed to reduce bat arrousals in all the monitoring periods, with a reduced number of field surveyors per site and a short observation time frame near the animals.

### Data collection

The study was performed in Romania, a country with diverse karst landscapes and large cave-dwelling bat populations [54]. We chose ten SEs (Fig 1, Table 1) located in the steppe (n = 3), continental (n = 1), and alpine bioregions (n = 6). We classified them into three categories, based on their origin and current use: wild caves, show caves, and mines.

The number of bats in colonies was determined using flash photography to minimize temperature-based arousals [56]. Observations were limited to the extent of the show cave paths, where most bats roosted (SEs 4, 5, 6, and 10).

Data collection was split into three periods: hibernation start (SO: September–October), mid to end hibernation (NM: November–March), maternity start (AM: April–May). Each period was sampled three times per year (beginning, mid, end), and each SE was observed for two consecutive years, between 2010 and 2016 (Table 1). The 2014–2016 period did not cover the complete maternity interval to minimize human impact on bat populations. The surveys from this period were useful for model development, as bats already form the colonies at the start of the maternity period.

The spatial extent of the studied SEs, which represented the modelling boundary, was modified from published maps (Fig 2), but we also performed new surveys where the data was outdated or inexistent (SE 1–3). We used a laser meter (Leica DistoX) fitted with an inclinometer and PDA—Paperless Cave Survey software [57, 58] for all cave surveys. The SEs which had published maps were also surveyed, but only using a central line which helped us georeference the existing maps. The survey data was processed using Compass Cave Survey and Mapping Software. We merged the final maps into a single layout, but each SE had its own spatial reference. This was used only as a visual aid for results presentation, while the models were performed at their original scale of all SEs.

The focal bat species were *Rhinolophus ferrumequinum*, *R, hipposideros*, *R. euryale*, *Myotis myotis* / *M. blythii*, and *Miniopterus schreibersii*. It was impossible to distinguish between colonies of *M. myotis* and *M. blythii* without causing arousal, and therefore they were treated as a single group. Less frequent species, such as *R. mehelyi*, *Nyctalus noctula*, and *Pipistrellus pipsitrellus* (Table 2), were also recorded.

Mapping bat occurrences and human disturbances within the SEs were performed using the same survey method. We created a central survey line, discretely marking stations for future reference. We then used these stations to determine the location of the occurrences. We recorded air temperatures close to the animals each time we collected occurrences. Supplementary data include the height at which bats were located relative to the floor and the distance from the nearest entrance (spot variables). The distance was extracted via Network Analyst

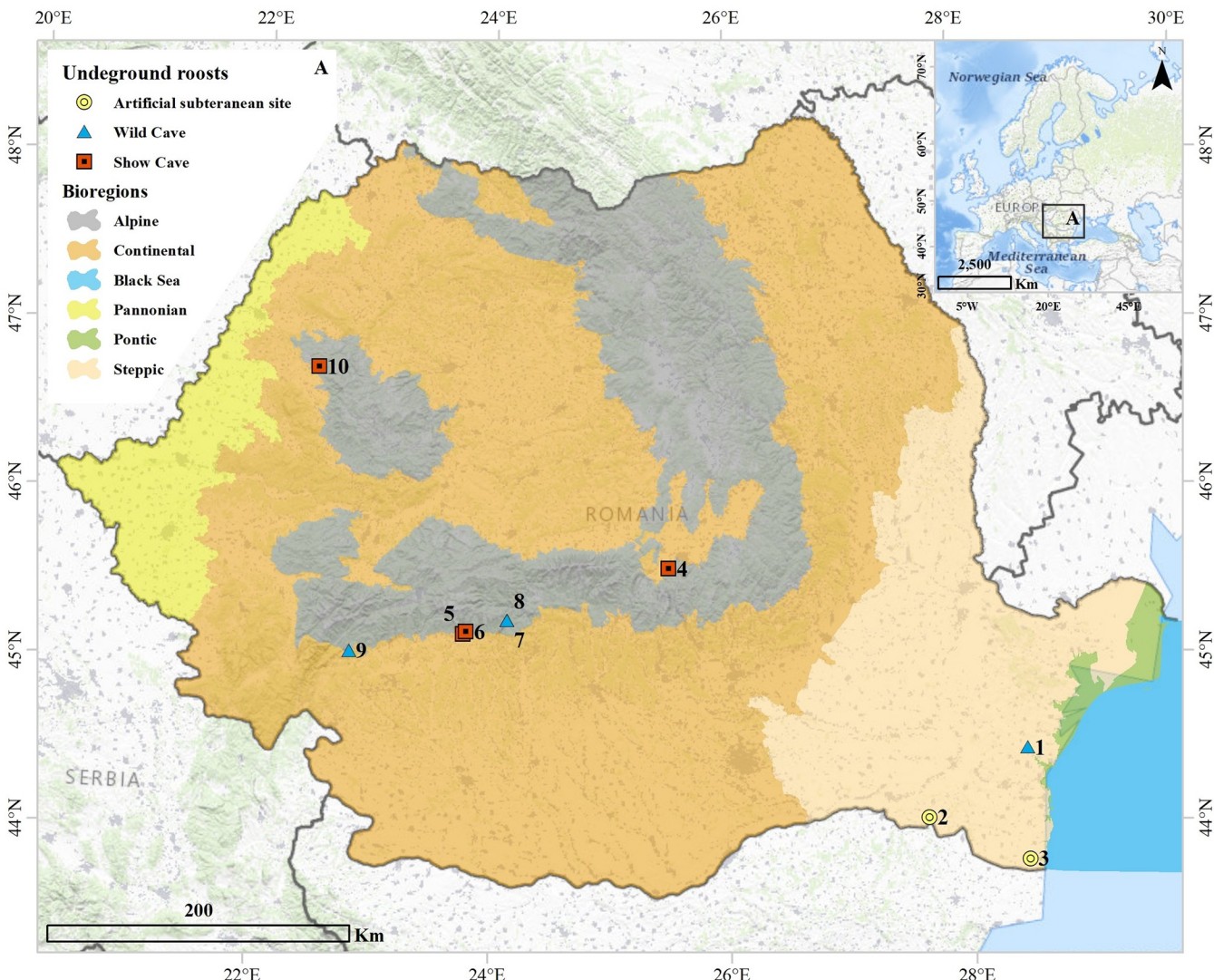

Service Layer Credits: USGS The National Map: National Boundaries Dataset, 3DEP Elevation Program, Geographic Names Information System, National Hydrography Dataset, National Land Cover Database, National Structures Dataset, and National Transportation Dataset; USGS Global Ecosystems; U.S. Census Bureau TIGER/Line data; USFS Road Data; Natural Earth Data; U.S. Department of State Humanitarian Information Unit; and NOAA National Centers for Environmental Information, U.S. Coastal Relief Model. Data refreshed June, 2022.

Biogeograpgic regions: EEA standard re-use policy: unless otherwise indicated, re-use of content on the EEA website for commercial or non-commercial purposes is permitted free of charge, provided that the source is acknowledged (https://www.eea.europa.eu/legal/copyright). Copyright holder: European Environment Agency (EEA).

**Fig 1. Locations of surveyed SEs, classified by type and biogeographical region [55].** 1—Gura Dobrogei cave, 2—Canaraua tunnel, 3—Hagieni tunnel, 4—Valea Cetății cave, 5—Muierilor cave, 6—Polovragi cave, 7—Stogu cave, 8—Lacul Verde cave, 9—Cloşani cave, 10—Meziad cave.

extension (ESRI). Human activity within show caves (SEs 4, 5, 6, 10) and a closed-circuit cave (SE 9) was measured with the help of people counters using a method described in [64, 65]. We used the number of passes recorded by people counters (tourist passes) as a proxy for the number of tourists who visited the caves. This proxy does not reflect the number of tourists because people counters log each movement regardless of direction. Non-touristic SEs were not monitored using this method, but we attributed 100 human passes per year for scalability. The resulting spatial datasets were processed within the ArcGIS (ESRI) environment.

**Table 1. Details regarding the studied subterranean environments.**

| Code | Name | Analysed period | Cavity type | Altitude (a.s.l.) m | No. Levels | Cave length (m) | Touristic length (m) | Tourist passes No. / year—2015 |
|------|------|-----------------|-------------|---------------------|-----------|-----------------|----------------------|-------------------------------|
| SE 1 | Gura Dobrogei cave* | 2012–2014 | Wild cave | 49 | 3 | 451 | | 100 |
| SE 2 | Canara Tunnel* | 2012–2014 | Artificial | 36 | 1 | 282 | | 100 |
| SE 3 | Hagieni Tunnel* | 2012–2014 | Artificial | 21 | 1 | 150 | | 100 |
| SE 4 | Stogu cave | 2010–2012 | Wild cave | 939 | 2 | 240 | | 100 |
| SE 5 | Lacul Verde cave | 2010–2012 | Wild cave | 992 | 2 | 183 | | 100 |
| SE 6 | Polovragi cave* | 2014–2016 | Show cave | 630 | 3 | 10350 | 500 | 52835 |
| SE 7 | Muierilor cave* | 2014–2016 | Show cave | 645 | 3 | 8000 | 800 | 407636 |
| SE 8 | Closani cave* | 2014–2016 | Wild cave | 433 | 2 | 1458 | | 100 |
| SE 9 | Valea Cetatii cave | 2014–2016 | Show cave | 825 | 2 | 958 | 120 | 22135 |
| SE 10 | Meziad cave* | 2014–2016 | Show cave | 435 | 2 | 6000 | 1000 | 14266 |

*Caves included within the EUROBATS important undergrounds site list

### Air temperature monitoring and environmental datasets

We measured hourly air temperature values using two methods: continuous monitoring via data loggers and spot temperature measurements with instant probes, as described in [65]. The time frame of the temperature monitoring study overlapped the species observation intervals. The climatic monitoring points were positioned in the field using the above survey method. The continuous temperature measurements were performed using Gemini Tinytag Plus 2 loggers (±0.01˚C accuracy, 0.02˚C resolution) and Hobo Pendant Temperature loggers (±0.53˚C accuracy, 0.14˚C resolution). Before the release of these data loggers, we used iButton Hygrocron from Maxim Integrated (±0.5˚C accuracy, 0.0625˚C resolution with post-processing corrections) for half of the sites (SEs 1, 2, 3, 7, 8). The loggers were positioned 2 m above the cave floor, in the central area of the passage section or chamber. Spot air temperature measurements were collected via a Vaisala HMP70 temperature probe (±0.2˚C accuracy, 0.01˚C resolution), both close to isolated bats or colonies and at specific points where site morphology could influence air temperature values. Spot measurements were later used to extract statistical data regarding bat preferences for an independent validation dataset in the temperature analysis. External air temperature recordings were collected near the SEs entrances, using meteorological stations (SE 5, 6, 9) or data loggers.

Cave air temperature surfaces were interpolated (0.5 m resolution) using the data logger information (S1 Table in S1 File) via the natural neighbour method [66]. Cave walls were used as boundaries. We used a non-parametric interpolation, opposed to the geostatistical approaches previously used, such as kriging [43], because the geometry of most SEs does not allow for a gridded sampling strategy. Daily interpolations were created, and then averaged to capture temperature variations for each relevant bat activity period. The observations were performed in the middle of the galleries; however, to interpolate a valid surface, each point was copied two times, placing the additional points perpendicular to the cave walls outside their boundaries. Given the fact that Meziad show cave (SE 10) stretches on two main levels (LVL1 and LVL2), which slightly overlap but have different microclimatic regimes, we analysed them as two distinct SEs. The interpolated results were merged into one layer, prioritizing the lower level because it held most of the collected occurrences.

Two validation methods (dependent and independent) were used to check the errors of the interpolations. This was achieved using cross-validation with the interpolation dataset, created

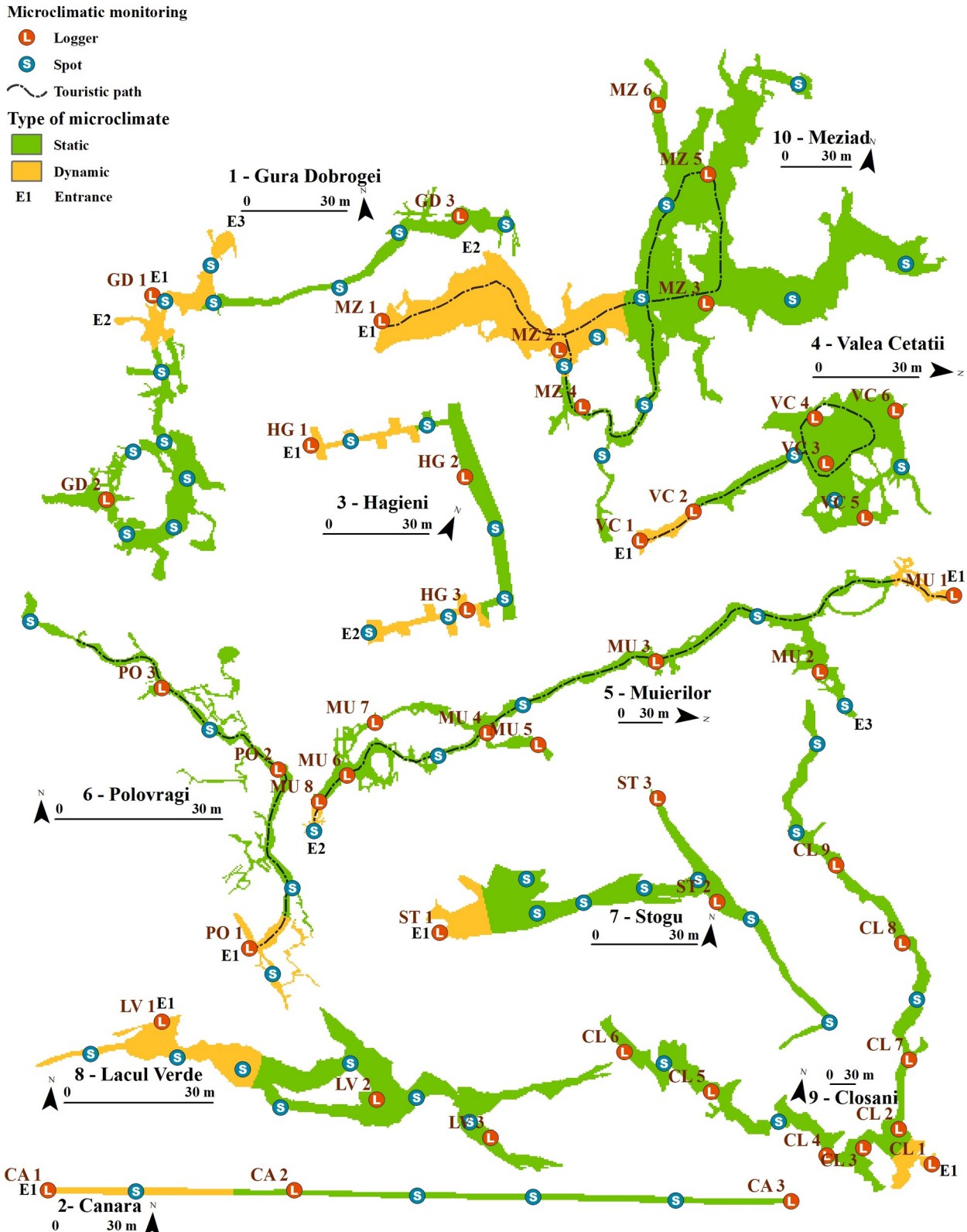

**Fig 2. Map of the selected SEs, location of the air temperature monitoring stations or spot measurements, and a classification of the yearly air temperature ranges: Static climate = < 5˚C yearly variations, dynamic climate = > 5˚C yearly variations.** Source of the georeferenced maps: Polovragi cave [59]; Cloşani cave [60]; Valea Cetăţii cave [61]; Lacul Verde, and Stogu caves [62]; Meziad cave [63].

**Table 2. Occurrences per species detailed by SE, activity period, and a maximum number of tourists' passes per bat activity period for the 2014–2016 observation interval.**

| Code. | Site | Species | Activity period | | | | | | | | |
|---|---|---|---|---|---|---|---|---|---|---|---|
| | | | Total no. of observations | | | Max no. of observations/ field visit | | | Max no. of tourist passes | | |
| | | | SO | NM | AM | SO | NM | AM | SO | NM | AM |
| SE 1 | Gura Dobrogei cave | *Miniopterus schreibersii* | 49 | 29 | 504 | 49 | 26 | 504 | 100 | 100 | 100 |
| | | *Myotis myotis/blythii* | 7 | 338 | 0 | 7 | 87 | 0 | 100 | 100 | 100 |
| | | *Rhinolophus ferrumequinum* | 13 | 223 | 1 | 13 | 45 | 1 | 100 | 100 | 100 |
| | | *Rhinolophus mehelyi* | 0 | 2 | 0 | 0 | 1 | 0 | 100 | 100 | 100 |
| SE 2 | Canara Tunnel | *Miniopterus schreibersii* | 0 | 338 | 0 | 0 | 338 | 0 | 100 | 100 | 100 |
| | | *Myotis myotis/blythii* | 0 | 5 | 0 | 0 | 3 | 0 | 100 | 100 | 100 |
| | | *Rhinolophus ferrumequinum* | 0 | 118 | 0 | 0 | 58 | 0 | 100 | 100 | 100 |
| | | *Rhinolophus hipposideros* | 0 | 4 | 0 | 0 | 2 | 0 | 100 | 100 | 100 |
| | | *Rhinolophus mehelyi* | 0 | 7 | 0 | 0 | 6 | 0 | 100 | 100 | 100 |
| SE 3 | Hagieni Tunnel | *Miniopterus schreibersii* | 0 | 449 | 0 | 0 | 448 | 0 | 100 | 100 | 100 |
| SE 4 | Valea Cetății cave | *Myotis myotis/blythii* | 0 | 8 | 0 | 0 | 6 | 0 | 942 | 21941 | 585 |
| | | *Rhinolophus ferrumequinum* | 2 | 22 | 0 | 2 | 11 | 0 | 942 | 21941 | 585 |
| | | *Rhinolophus hipposideros* | 3 | 5 | 0 | 3 | 3 | 0 | 942 | 21941 | 585 |
| SE 5 | Muierilor cave | *Miniopterus schreibersii* | 1363 | 929 | 0 | 1363 | 883 | 0 | 11856 | 66459 | 9740 |
| | | *Myotis myotis/blythii* | 11 | 7 | 0 | 11 | 5 | 0 | 11856 | 66459 | 9740 |
| | | *Rhinolophus ferrumequinum* | 875 | 2061 | 70 | 875 | 1112 | 70 | 11856 | 66459 | 9740 |
| | | *Rhinolophus hipposideros* | 20 | 82 | 1 | 20 | 49 | 1 | 11856 | 66459 | 9740 |
| SE 6 | Polovragi cave | *Miniopterus schreibersii* | 0 | 7 | 0 | 0 | 7 | 0 | 3032 | 5330 | 10929 |
| | | *Myotis myotis/blythii* | 5 | 18 | 3 | 5 | 13 | 3 | 3032 | 5330 | 10929 |
| | | *Rhinolophus ferrumequinum* | 416 | 542 | 0 | 416 | 283 | 0 | 3032 | 5330 | 10929 |
| | | *Rhinolophus hipposideros* | 11 | 29 | 1 | 11 | 16 | 1 | 3032 | 5330 | 10929 |
| SE 7 | Stogu cave | *Myotis myotis/blythii* | 0 | 3 | 0 | 0 | 3 | 0 | 100 | 100 | 100 |
| | | *Rhinolophus ferrumequinum* | 0 | 28 | 9 | 0 | 28 | 9 | 100 | 100 | 100 |
| | | *Rhinolophus hipposideros* | 0 | 7 | 3 | 0 | 7 | 3 | 100 | 100 | 100 |
| SE 8 | Lacul Verde cave | *Myotis myotis/blythii* | 0 | 3 | 7 | 0 | 3 | 7 | 100 | 100 | 100 |
| | | *Rhinolophus ferrumequinum* | 0 | 34 | 33 | 0 | 34 | 33 | 100 | 100 | 100 |
| | | *Rhinolophus hipposideros* | 0 | 4 | 1 | 0 | 4 | 1 | 100 | 100 | 100 |
| SE 9 | Closani cave | *Rhinolophus euryale* | 246 | 342 | 22 | 246 | 255 | 22 | 100 | 100 | 100 |
| | | *Rhinolophus ferrumequinum* | 21 | 63 | 0 | 21 | 15 | 0 | 100 | 100 | 100 |
| | | *Rhinolophus hipposideros* | 0 | 1 | 0 | 0 | 1 | 0 | 100 | 100 | 100 |
| SE 10 | Meziad cave Level 1 | *Miniopterus schreibersii* | 2 | 2002 | 384 | 2 | 1230 | 384 | 2778 | 1334 | 2531 |
| | | *Myotis myotis/blythii* | 3 | 4 | 0 | 3 | 3 | 0 | 2778 | 1334 | 2531 |
| | | *Nyctalus noctula* | 0 | 1 | 0 | 0 | 1 | 0 | 2778 | 1334 | 2531 |
| | | *Pipistrellus pipistrellus* | 230 | 1 | 0 | 230 | 1 | 0 | 2778 | 1334 | 2531 |
| | | *Rhinolophus ferrumequinum* | 264 | 504 | 1 | 264 | 281 | 1 | 2778 | 1334 | 2531 |
| | | *Rhinolophus hipposideros* | 3 | 138 | 2 | 3 | 94 | 2 | 2778 | 1334 | 2531 |
| | Meziad cave Level 2 | *Miniopterus schreibersii* | 0 | 0 | 1 | 0 | 0 | 1 | 2778 | 1334 | 2531 |
| | | *Myotis myotis/blythii* | 0 | 2 | 279 | 0 | 1 | 279 | 2778 | 1334 | 2531 |
| | | *Nyctalus noctula* | 0 | 42 | 0 | 0 | 42 | 0 | 2778 | 1334 | 2531 |
| | | *Rhinolophus ferrumequinum* | 27 | 7 | 1 | 27 | 4 | 1 | 2778 | 1334 | 2531 |
| | | *Rhinolophus hipposideros* | 3 | 1 | 10 | 3 | 1 | 10 | 2778 | 1334 | 2531 |
| **Total** | | | **3574** | **8410** | **1333** | **3574** | **5410** | **1333** | | | |

from the data logger measurements (dependent validation, S2 Table in S1 File) and a separate dataset using the spot variables (independent validation). We used Mean Absolute Error (MAE) and Root Mean Squared Error (RMSE) to compare the results, where lower values are considered to show better results and equal values show errors in the same magnitude [67].

We included data on human-induced temperature changes (resulting from tourist presence and light systems) collected from show caves used in our study [65].

## Statistical analysis

To assess whether there are statistically significant differences between the total number of bats from all the SE over the three analysed periods, the number of individuals from each species during the activity periods, and the total number of bats per each SE over the three activity periods, we performed a series of Friedman's tests. The effect size of the test was measured with Kendall's W coefficient. Pairwise comparisons using paired Wilcoxon signed-rank tests were used to identify the activity periods with significant differences. We, furthermore, calculated the Spearman rank correlation coefficient to estimate the measure of association between the cave length and size of bat populations and also between the cave length and species diversity during the analysed activity periods.

We used Generalised Linear Models (GLM) with stepwise backward elimination to evaluate the differences between five bat species (*Rhinolophus euryale*, *R. hipposideros*, *R. ferrumequinum*, *Miniopterus schreibersii*, and *Myotis myotis / blythii*) in relation to torpor height relative to the SE floor, the distance from the entrance, and measured air temperature for all the monitoring periods. We performed a GLM with a binomial distribution using predictor variables for each species. We tested the variable multicollinearity using the Variance Inflation Factors (VIF) function in R, eliminating values greater than 10 [68]. The best model selection was achieved using the lowest Akaike's Information Criterion (AIC) value. To identify the relation between bat abundance per species and periods, compared to the number of people who visited the show caves, and for comparisons of cave metrics and bat populations, we performed Spearman's correlation tests. The analyses were performed in R statistical software version 3.6.3 with the "stats" package [69].

## Species distribution modelling

We classified occurrences per species into three periods, as mentioned above, and excluded species with less than 25 occurrences [70]. Spatial autocorrelation revealed heavily clustered datasets (Moran's I index = 0.72, z = 926.14, P ≤ 0.001) because bats form clustered colonies to conserve energy [24]. To account for the autocorrelation errors caused by the species' clustering during torpor or maternity, we have chosen to model the distribution of each species in all SEs for each activity period with no spatial thinning, increasing model variability. Some bat species were not found in all of the analysed SEs (Table 2). Our models were calibrated for each species and selected period within the geometry and environmental datasets of all SEs.

We used ESDM (Ensemble Species Distribution Models) from the SSDM (Stacked Species Distribution Models) R package [71], generating an ensemble model from multiple algorithms: classification tree analysis, multivariate adaptive regression splines, generalized linear models, artificial neural networks, general additive models, support vector machines, and Maxent. The ensemble method generates robust outputs using a large set of algorithms, improving the accuracy of the predictions [51]. These were stacked together into one AUC weighted projection. Models were calibrated with a random subset extracted from the occurrence data (75%) and evaluated with the remaining 25% test datasets [72, 73]. The variable contribution was calculated using the Pearson correlation coefficient, SSDM R package. The performance of the

models was evaluated using the Receiver Operating Characteristic (ROC) Area Under the Curve (AUC) [74] and Cohen's K test [75]. Both range from 0 to 1, and the maximum value indicates the best fit of the model. In addition, we used a ten-percentile omission error, obtained with the method described by Gherghel and Martin [76], because it is an independent metric extracted from the validation dataset and shows how well the model discriminates between presence and absence predictions. We summed the final predictions using ArcGIS (ESRI) cell statistics into separate datasets per species and period and further summed these into a cumulative dataset for each SE with no prioritization, considering that all bat species are equally sensitive to human disturbance.

## Results

### Data collection

SEs varied from large wild caves with complex climatic systems (SEs 7, 10; Table 1) to small horizontal artificial sites (SEs 2, 3) with strong seasonal climatic shifts.

Bat distribution compared to the SE temperature range per activity period was plotted in the supplementary materials (S1-S3 Figs in S1 File). Most bats were found in more stable temperature ranges during hibernation and in more variable temperature ranges during the maternity period.

The five focal bat species were recorded in most of the analysed SEs, while the less frequent bat species were observed only in a few sites (Table 2).

### Air temperature monitoring and environmental datasets

Most of the SEs presented low and tolerable errors for bat activity (MAE < 0.4, RMSE < 0.4), except for Gura Dobrogei and Valea Cetății caves (SEs 1 and 4) which showed higher RMSE values (0.6 and 0.9, respectively). These were recorded in subterranean sectors which were located closer or between the entrances, but also where longitudinal profiles show vertical changes. Slight differences in accuracy and resolution of the equipment also contributed to some errors. The values were recorded in patches with low bat abundances and were insignificant compared to the dynamic variations recorded at the entrance of the SEs or the ecological requirements of bats, therefore, we considered the errors negligible. The independent validation also showed minor differences between the measured and the predicted values, with a maximum MAE of 0.4 and RMSE of 0.9 for Gura Dobrogei cave (SE 1), validating the interpolations.

Subterranean environments have a vertical temperature gradient, with hotter air concentrating closer to the ceiling. Due to field collection restrictions, this was not covered by the data logger monitoring method, especially in high chambers. Nevertheless, some spot measurements were collected at much greater heights, matching the height of the bats. Differences are shown in the independent test datasets (S2 Table in S1 File), but they do not exceed the tolerable errors relevant for torpid or nursing bats (< 0.9˚C).

Some temperature disturbances were recorded within the show caves, with an increase by 0.4˚C near the lights in Polovragi cave (SE6). These appeared during the visiting period and were proportional to the number of people recorded on the people counters. They reverted to the initial stable temperature (IST) in approximately 40 hours. Muierilor cave (SE 5) recorded higher temperature spikes in the touristic season (2˚C, over 250000 passes), which slowly reverted to IST in a month. Increased values of 0.5 to 1˚C were still recorded in medium-sized halls during the winter, where smaller groups of tourists were stationed. During winter, the number of tourists dropped; therefore, temperature values reverted faster to the IST (one day).

Valea Cetății and Meziad caves (SEs 4 and 10) showed minimal temperature variations associated with human disturbances.

## Statistical analysis based on spot measurements and observations

Friedman's test showed that the number of bats was significantly different during the three analyzed periods (P < 0.05), with a moderate effect size (W = 0.44). The results of the Wilcoxon signed-rank test showed significant pairwise differences between NM–SO (P < 0.05) and NM–AM (P < 0.05). While testing for differences between the number of individuals from each species and activity periods, the results of Friedman's test showed significance for *Myotis myotis/Myotis blythii* (P = 0.0368), *Rhinolophus ferrumequinum* (P = 0.0003), and *R. hipposideros* (P = 0.0129). The following SEs registered significant differences among the number of bats over the three activity periods: Muierilor cave (SE 5: P = 0.0498, W = 0.75), Polovragi cave (SE 6: P = 0.0224, W = 0.95), and Canara Tunnel (SE 2: P = 0.0067, W = 1). At the beginning of the hibernation period (SO), the cave length was strongly correlated with the size of the bat populations (Spearman's rho = 0.9192, P < 0.005), but this correlation did not follow during the NM and AM periods. Species diversity also correlated positively with cave length during SO (Spearman's rho = 0.84966, P < 0.005), but not during NM and AM.

Results of the GLM showed that the species were significantly influenced by height, distance from the cave entrance, and measured temperature (spot variables–S3 Table in S1 File) during at least one of the analysed activity periods (Table 3).

At the beginning of hibernation (SO), *R. hipposideros*, *R. ferrumequinum*, *M. schreibersii*, and *M. myotis / blythii* showed a preference for the higher temperatures, closer distances from the entrance (except for *M. myotis / blythii*), and higher hibernation heights (only *R. euryale* and *M. schreibersii*). In contrast, during the mid-end part of the hibernation (NM), the species preferred lower heights, greater distances from the entrances, and lower temperatures. During maternity (AM), all species except *R. ferrumequinum* searched for higher temperatures, while *R. ferrumequinum* and *M. myotis / blythii* also showed a preference for greater heights relative to the SE floor (Table 3).

Correlation between bat abundance per species and observation periods compared to the number of tourists' passes in show caves showed insignificant values, except for *R. hipposideros* during the start of the hibernation (Table 4).

Thermophilic species such as *R. mehelyi* and *R. euryale* searched for warmer areas during hibernation compared to crevice-dwelling species, such as *Nyctalus* sp. or *Pipistrellus* sp. Most *Rhinolophus* spp. choose roosts with lower temperature variations and preferably one entrance. *Rhinolophus* spp. hibernated for more extended periods than crevice-dwelling species, which preferred cracks in large chambers (in SE 10), with higher temperature amplitude. Distance from the entrances increased when exterior temperature decreased, except for *M. myotis / blythii*, which got closer to the entrance in those conditions. Most species decreased their roosting height towards the end of the hibernation in search of lower temperatures to conserve energy. In contrast, greater heights were linked to warmer areas optimal for nursing during maternity for all the studied species.

## Species distribution models

Results showed potential suitable habitats for most of the focal species in most SEs, but not all species have yet to colonize those areas. Ensemble models performed well, with a minimum of 0.75 and an average of 0.89 training AUC. Cohen's Kappa also showed optimal averaged results, reaching 0.81. Suitable habitats per species were accurately predicted, with an averaged 10% omission rate of 8.3%. The selected model variables changed for each activity period

**Table 3. Linear regression per species, activity periods, and spot variables.**

| Species | Spot variables | Hibernation start (SO) | | Hibernation mid-end (NM) | | Maternity (AM) | |
|---|---|---|---|---|---|---|---|
| | | t value | Pr (>\|t\|) | t value | Pr (>\|t\|) | t value | Pr (>\|t\|) |
| *Rhinolophus Euryale* | Height | **13.45** | **<0.05** | **-9.91** | **<0.05** | - | - |
| | Distance | - | - | **-5.50** | **<0.05** | **2.85** | **<0.05** |
| | Temperature | **-11.78** | **<0.05** | -0.7 | 0.48 | **5.11** | **<0.05** |
| *Rhinolophus hipposideros* | Height | | - | -1.44 | 0.14 | - | - |
| | Distance | **-4.90** | **<0.05** | **4.34** | **<0.05** | 1.40 | 0.16 |
| | Temperature | **3.45** | **<0.05** | **-7.43** | **<0.05** | **4.52** | **<0.05** |
| *Rhinolophus ferrumequinum* | Height | **-11.59** | **<0.05** | 1.87 | 0.06 | **14.40** | **<0.05** |
| | Distance | **-9.87** | **<0.05** | **6.01** | **<0.05** | **2.66** | **<0.05** |
| | Temperature | **28.15** | **<0.05** | **-27.96** | **<0.05** | **-2.28** | **<0.05** |
| *Miniopterus schreibersii* | Height | **11.28** | **<0.05** | **-4.11** | **<0.05** | **-7.96** | **<0.05** |
| | Distance | **-15.54** | **<0.05** | **7.67** | **<0.05** | **-2.24** | **<0.05** |
| | Temperature | **22.71** | **<0.05** | **-22.50** | **<0.05** | **24.74** | **<0.05** |
| *Myotis myotis / blythii* | Height | **-6.46** | **<0.05** | **-3.73** | **<0.05** | **6.12** | **<0.05** |
| | Distance | **3.94** | **<0.05** | **-7.08** | **<0.05** | **4.78** | **<0.05** |
| | Temperature | **2.44** | **<0.05** | **-4.97** | **<0.05** | **2.99** | **<0.05** |

Bold = statistically significant results.

according to each species' environmental requirements. During the start of hibernation, minimum temperatures were more relevant; during the mid-end hibernation, the maximum temperatures and the temperature ranges were significant, while during the maternity period, the minimum and maximum temperatures were mainly considered (Table 5).

The sum of the binary models per period (Figs 3–5) showed where an aggregation of species was more likely to appear. During SO, the models showed a higher abundance of optimal habitats in areas that mostly overlapped the static temperature sectors of the SEs. Colder SEs presented higher SDM values, such as SE 4, SE 7, and SE 8. These favourable sectors became more restrictive during NM and concentrated at lower stable temperatures than SO, but with some degree of temperature variability. The colder SEs lost some degree of favourability compared to the more complex roosts, such as SE 5, SE 9, and SE 10, where the animals could find optimal climates for torpor. The AM period showed substantial bat abundance and distribution changes, with essential sectors closer to the entrance. Meziad cave (SE 10) showed a climatic inversion, where hot air reached and accumulated in the upper level, at the far end of the show cave sector. Nevertheless, the SDMs accurately predicted these important roosting areas for bats (*M. myotis*). Cold caves such as Valea Cetății or Stogu (SE 4, SE7, respectively) had little to no predicted SDMs during AM.

**Table 4. Spearman's correlation matrix, comparing species abundance per roost with the number of tourists' passes per activity period.**

| Species | Hibernation Start (SO) | | Hibernation Mid-end (NM) | | Maternity (AM) | |
|---|---|---|---|---|---|---|
| | rho | P-value | rho | P-value | rho | P-value |
| *Miniopterys schreibersii* | 0.41 | 0.40 | 0.21 | 0.68 | -0.28 | 0.58 |
| *Myotis myotis/ blythyi* | 0.61 | 0.10 | 0.44 | 0.27 | 0.29 | 0.48 |
| *Rhinolophus ferrumequinum* | 0.81 | **<0.05** | 0.38 | 0.30 | 0.02 | 0.94 |
| *Rhinolophus hipposideros* | **1** | **<0.05** | 0.53 | 0.17 | 0.29 | 0.48 |
| (Total) No. of bats vs. no. of people | **0.60** | **<0.05** | 0.19 | 0.29 | 0.09 | 0.61 |

**Table 5. SDM results per species—ensemble model evaluation and variable importance contribution.**

| SDMs | | Model evaluation | | | Variable importance | | | | |
|---|---|---|---|---|---|---|---|---|---|
| Period | Species | AUC | Omission Rate | Cohen's Kappa | Tmax | Tmean | Tmin | Trange | Tstd |
| SO | *Miniopterus schreibersii* | 0.95 | 0.03 | 0.90 | | | 74.29 | 13.88 | 11.83 |
| SO | *Myotis myotis/blythii* | 0.81 | 0.06 | 0.71 | | | 62.41 | 37.59 | |
| SO | *Pipistrellus pipistrellus* | 0.97 | 0.00 | 0.93 | 28.78 | | | | 71.22 |
| SO | *Rhinolophus euryale* | 0.95 | 0.01 | 0.90 | 63.06 | | | | 36.94 |
| SO | *Rhinolophus ferrumequinum* | 0.88 | 0.03 | 0.75 | | | 60.66 | 39.34 | |
| SO | *Rhinolophus hipposideros* | 0.75 | 0.07 | 0.72 | | | 73.79 | | 26.21 |
| NM | *Miniopterus schreibersii* | 0.87 | 0.07 | 0.75 | 32.99 | 34.36 | | 32.65 | |
| NM | *Myotis myotis/blythii* | 0.87 | 0.08 | 0.74 | | 46.17 | | 53.83 | |
| NM | *Nyctalus noctula* | 0.93 | 0.04 | 0.87 | 47.38 | 52.62 | | | |
| NM | *Rhinolophus euryale* | 0.93 | 0.02 | 0.86 | 65.22 | | | 34.78 | |
| NM | *Rhinolophus ferrumequinum* | 0.84 | 0.06 | 0.70 | 31.81 | | 68.19 | | |
| NM | *Rhinolophus hipposideros* | 0.80 | 0.09 | 0.73 | 32.13 | | 67.87 | | |
| AM | *Miniopterus schreibersii* | 0.92 | 0.05 | 0.83 | 41.83 | | 58.17 | | |
| AM | *Myotis myotis/blythii* | 0.97 | 0.01 | 0.95 | 39.32 | | 60.68 | | |
| AM | *Rhinolophus euryale* | 0.90 | 0.07 | 0.80 | 56.34 | | 43.66 | | |
| AM | *Rhinolophus ferrumequinum* | 0.96 | 0.06 | 0.91 | 50.47 | | 15.70 | | 33.82 |
| AM | *Rhinolophus hipposideros* | 0.86 | 0.08 | 0.72 | | | 43.23 | 30.30 | 26.48 |

## Discussions

This study emphasized the importance of using ensemble species distribution models on subterranean environments to predict suitable habitats for bat species during their lifecycle. For this novel method in terms of scale and functionality, we applied the existing ensemble species distribution modelling techniques on a group of temperate bat species within various caves and mines as a proof of concept. This approach is useful in conservation practices of cave-dwelling bats or any other subterranean biota [48], especially for show cave managers, as it highlights areas where anthropogenic impacts need to be minimised. The approach also offers a good insight regarding the ecology of cave-dwelling bats.

### Ecology of the focal bat species

During the cold season, food scarcity may drive temperate bats (especially cave-dwellers) into prolonged torpor [77]. The optimal torpor temperatures and torpor bout durations vary according to each species' biology and fitness [27], as bats choose roosting patches that can respond to exterior temperature changes and offer stimuli if the conditions are favourable for feeding, while keeping an optimal torpor temperature in unsuitable feeding conditions [27]. If temperatures decrease, the animals become aroused and change torpor locations [21]. For example, our analysis confirmed this by the temperature range and the maximum temperature variables, which had the highest model contribution towards the mid-end hibernation period, limiting optimal habitat patches within hotter and more stable temperatures for thermophile bat species, such as *Rhinolophus spp*. Roost length was also relevant for bat abundance and species diversity in all of our studied subterranean environments, as Torquetti et al. [19] also found, and was linked to larger patches of stable temperatures, therefore a wider selection of optimal habitats. At the beginning of the hibernation period most species and populations were corelated with smaller distances from the entrances, because they use the climatic variability to exit daily torpor and feed in optimal conditions [78], but during the colder winter

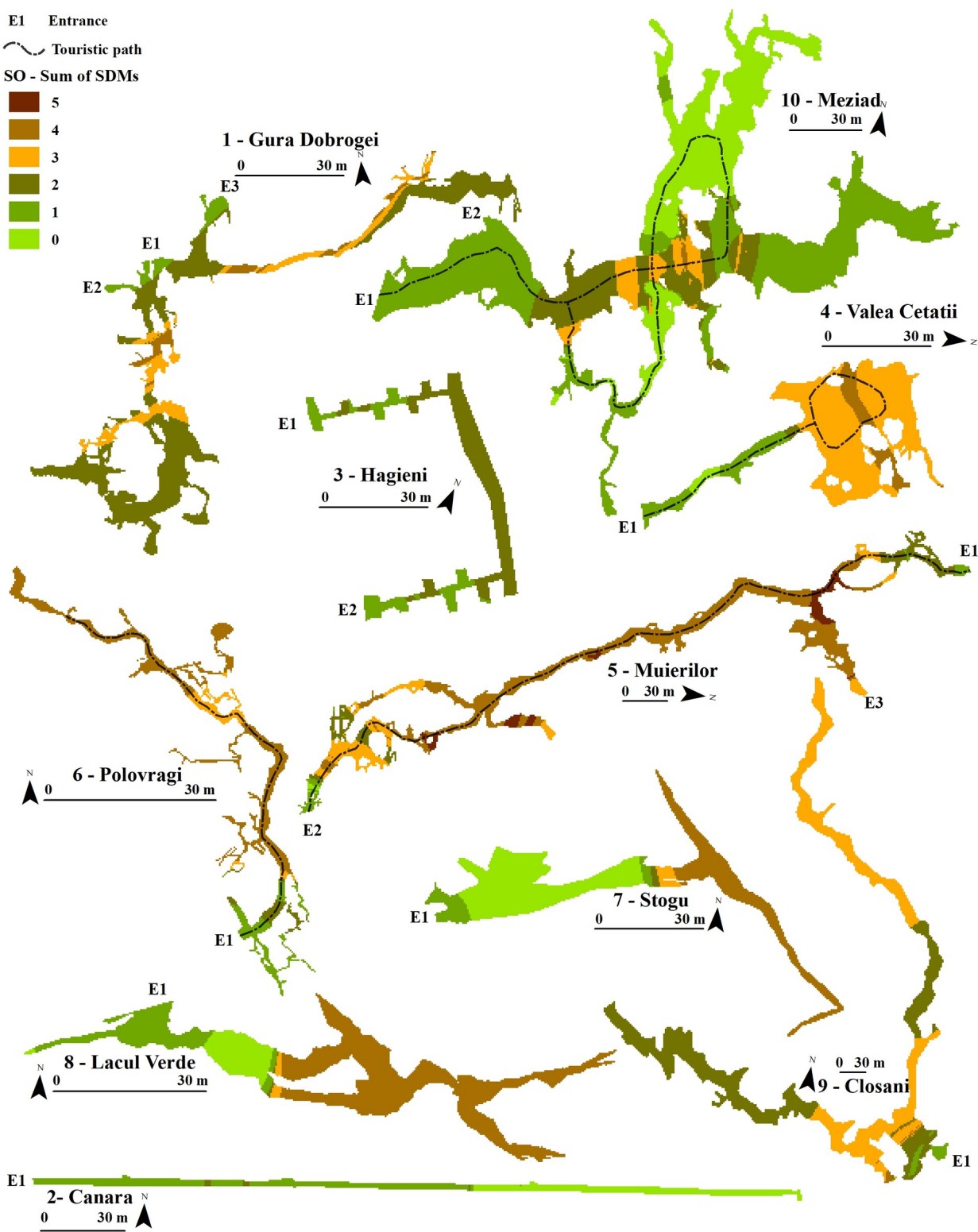

**Fig 3. Ensemble SDMs for the September–October—(SO) start of the hibernation period for all the studied SEs.** Sum of SDMs shows the number of suitable habitats which overlap within a SE.

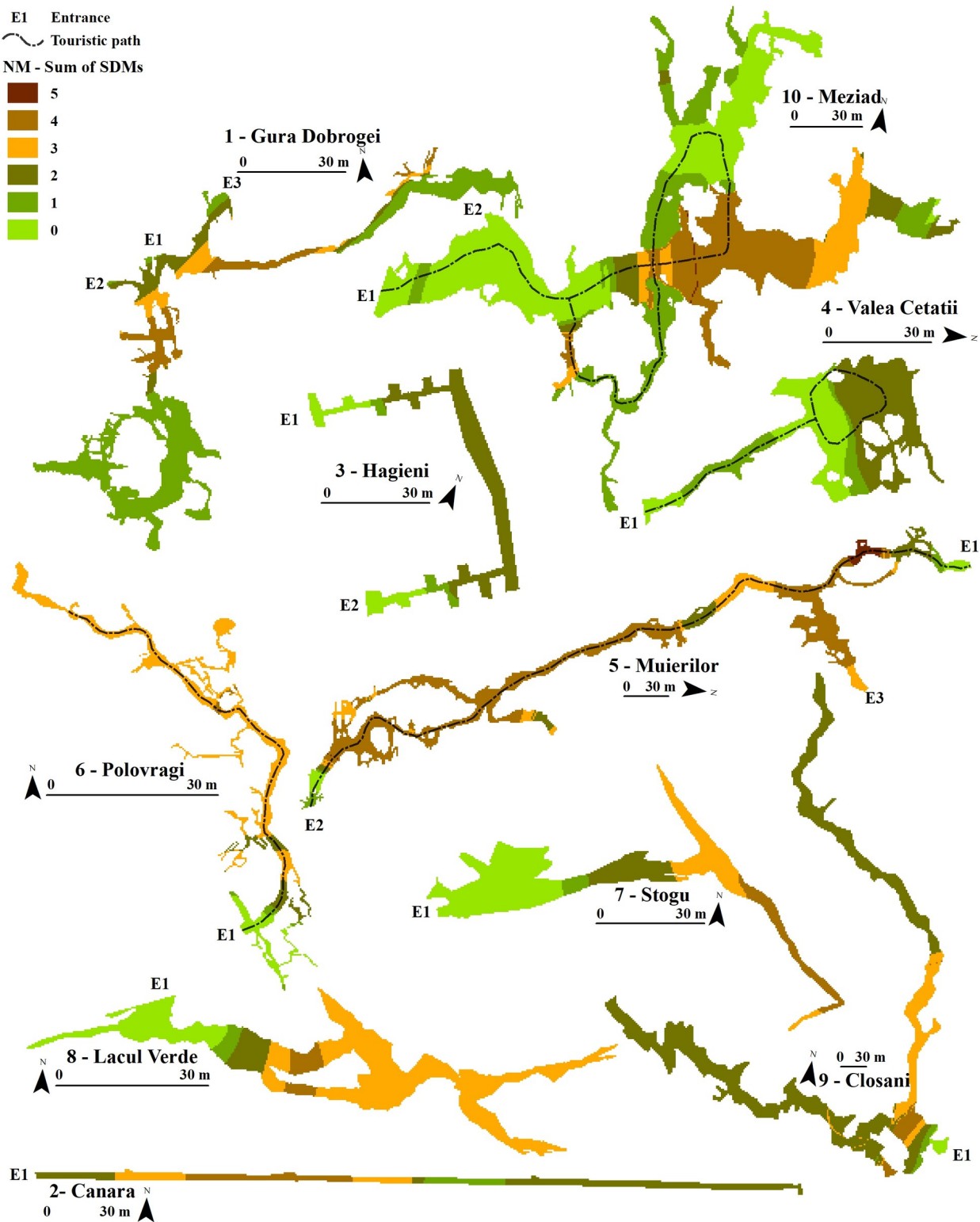

**Fig 4. Ensemble SDMs for the November–March (NM)—mid to end hibernation period for all the studied SEs.** Sum of SDMs shows the number of suitable habitats which overlap within a SE.

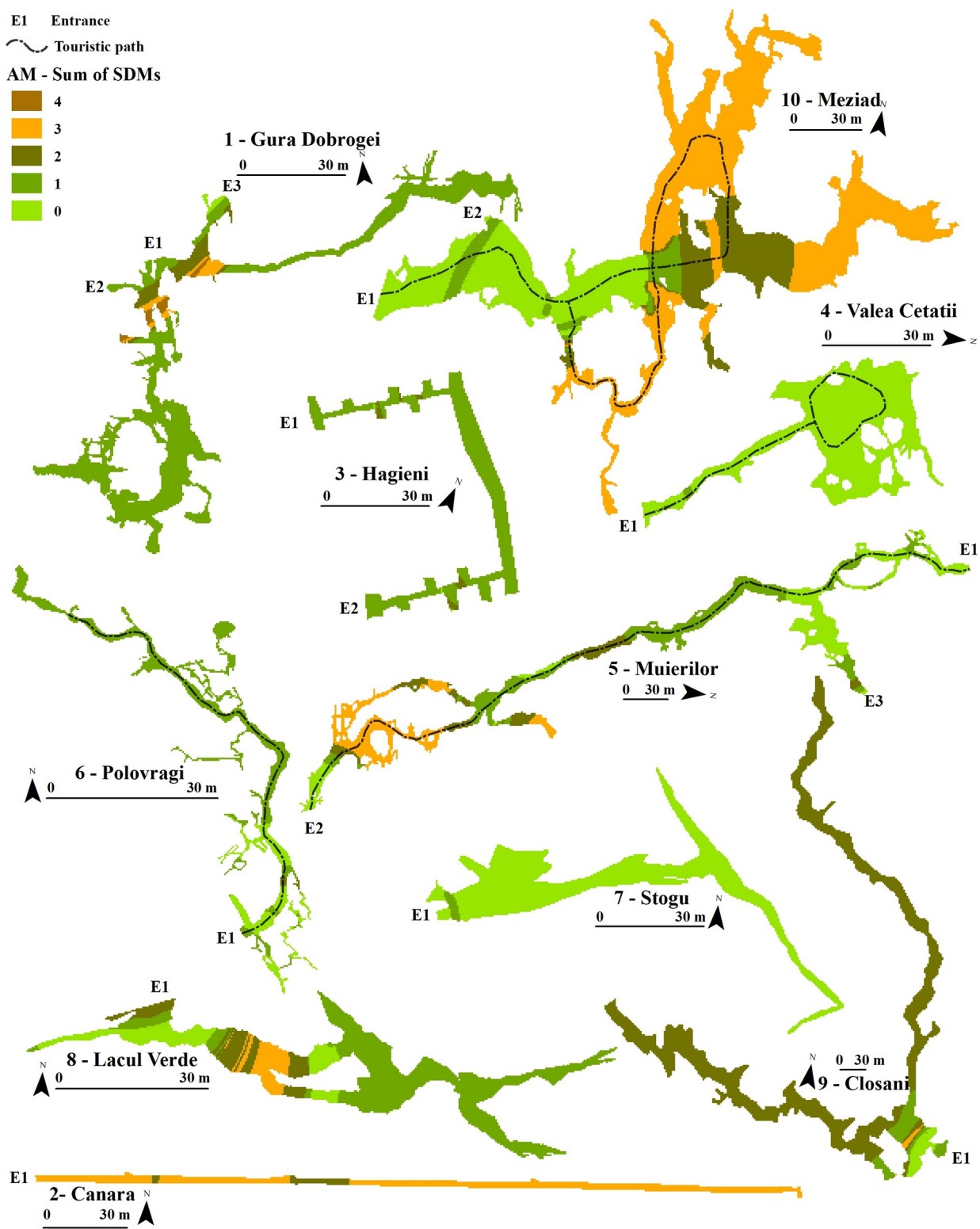

**Fig 5. Ensemble SDMs for the April–May (AM)—pre-maternity period for all the studied SEs.** Sum of SDMs shows the number of suitable habitats which overlap within a SE.

months these correlations were weaker suggesting that populations dispersed in various optimal habitat patches, according to their body fitness [23]. Nevertheless, a general movement towards the deeper sections of the cave was observed for species such as *Miniopterus schreibersii*, *Rhinolophus ferrumequinum* or *R. euryale*. These populations also decreased their roosting height in the mid-end hibernation period, which suggested that the animals were searching for colder temperatures to conserve energy [79]. During the end of the hibernation season, the animals increased their activity within the roosts and move closer to the entrances, starting to feed and disperse [80].

Because of their limited fat reserves, *R. hipposideros* individuals preferred lower hibernation heights, where colder air masses accumulated, also observed by Ruf & Geiser [23]. The species was also described as thermophile in caves located in its northern range [78], most likely due to a lower roost stable temperature. Torpid individuals were also much more dispersed compared to other studied *Rhinolophus* spp. Populations during hibernation, as a high degree of clustering can increase body temperature and cause arousals [23]. This makes them much more vulnerable to anthropogenic pressures, which can be more intensely applied in the lower part of a show cave vertical section.

On the opposite spectrum of these preferences, crevice-dwelling bats such as *Nyctalus noctula*, or *Pipistrellus pipistrellus* and some populations of *Myotis myotis / blythii* were not pushed within the deeper parts of the roosts during the coldest periods, with some examples where populations increased their proximity to the entrances. These species have shorter torpor bout durations compared to most cave-dwelling bat species and can optimally feed in low temperatures, thus proximity to the entrances is relevant in most conditions, so the animals can naturally be aroused in optimal feeding conditions [78].

The maternity period usually requires proximity to the entrances, where temperatures are more suitable for nursing [19], except for the case of temperature inversions, where the colonies can be formed even in deep sectors of the roosts (SE 10), as our models also predicted. Often smaller sites with exterior hot air influxes are preferred [19]. Roost sectors with higher annual temperature variations were more suitable for maternity colonies (SE 3) because hot air can warm the site, offering better conditions for nursing [81]. Warmer conditions during maternity can influence embryogenesis, keeping the females more active, but also ensuring warmer nursing conditions for pups during the early stages of their life [82].

Differences in the number of individuals for each subterranean environment and activity period suggest that some roosts are used as hibernaculum (especially the larger sites—SEs 5,10) while others as maternity (smaller sites—SEs 2 and 3), as bats perform seasonal migrations between these types of roosts [83].

## Species distribution models

Our approach has shown that modelling the habitat suitability of bat populations in subterranean environments is most likely less restrictive than previously considered [46], especially in large systems with more stable microclimates. External climatic factors are relevant for cave-dwelling bats, influencing their movements within the roosts [16], yet we have shown that they can be easely integrated within the environmental variables. Moreover, our models were able to predict seasonal level usage of the focal species within the studied environments, due to the accurate temperature interpolations coupled with a relatively narrow niche of optimal microclimatic conditions.

The focal bat species used open spaces within their roosts to hibernate or to raise their young, allowing for a standardised occurrence sampling strategy. Data collection within a controlled modelling environment, such as a cave or a mine, ensures a consistent sampling effort,

but only when dealing with open space dwellers [84]. Most troglobiontes, which make use of an extensive network of cracks and superficial habitats, would be more difficult to sample using this approach [46]. The method ensures a uniform sampling effort for most of the prediction area, offering a much better understanding of the results compared to regional or continental scale models [85].

While a smaller prediction area offers more control over the sampling strategy and the validation process, spatial autocorrelation may appear, especially when dealing with gregarious species which form colonies [79]. This can strongly influence model performance [86, 87] by introducing errors resembling oversampling. However, increasing the number of surveyed sites (hence increasing the variation in the background dataset from which the models can be trained) and performing the models in multiple subterranean environments simultaneously, helped us reduce this effect, while generating valid results. Applying occurrence rarefiation methods or spatial gridded sampling strategies on our dataset, as suggested by Mammola et al. [46], to account for spatial autocorrelation would have eliminated most of the collected occurrences, generating non-valid models, but this can be further explored in larger research efforts, and especially in large subterranean environments. Heavily clustered occurrences (e.g. data from a single small habitat patch) will, nevertheless, produce extremely restrictive models, which need to be validated before continuing the analysis.

Temperature was the main limiting factor for bat colonies in subterranean environments [16], thus, our models were solely based on this parameter and had optimal validated results. This modelling approach can be more difficult in large biomass caves, especially within the tropical or equatorial regions, because the presence of bat colonies creates hotter microclimates, influencing both the environmental variables and the model results, as Lundberg & Mcfarlane showed in a study focused on temperature disturbance generated by bats [44].

Bat ranges can have a high latitude variation resulting in geographical isolated populations which are exposed to different climatic conditions [88]. For example *Rhinolophus hipposideros* stretches from Northern Africa to the southern part of Great Britain. This will, in turn, influence their behaviour and climatic preferences in subterranean environments, with southern populations becoming more active during the cold season [88]. Thus, our method of modelling a single taxon over a wide range of sites can lead to strong overpredictions in studies with high roost latitude variations, and needs to be further adapted and tested to account for these conditions.

Although we have used an equal contribution sum for all of the modeled species within the final SDM results, considering they are euqally susceptible to human disturbance, a further development of this approach can include weighted overlays, in which species that are found to be more susceptible to anthropogenic pressure can be prioritised in the final sensibility maps.

## Implications for the conservation management of bats in subterranean environments

The models have showed abundances of favourable patches for bats in relation to their environments. We further compared these results with the anthropogenic impact to identify where conservation measures need to be applied.

Long-term anthropogenic impact can lead to roost abandonment [35], as shown in the Polovragi cave (SE 6). The cave hosted mixed maternity and hibernation colonies, but their numbers have dropped in recent years and the maternity colony has disappeared, most likely because of the upward-oriented lighting system and the human-bat interactions encouraged by the managers. The models showed sensible areas throughout the mid to end (deep) tourist

sectors. Thus, a more controlled touristic access in Polovragi (SE 6) cave is required to restore bat populations, with a new lighting system pointing downward and with a strict no human-bat interaction policy. The cave should not be visited during the hibernation season and restrictions should be imposed during maternity to increase the chances of bat recolonisation.

Large bat populations can still reside in sites with a high number of tourists [78], such as the Muierilor cave (SE 5). A stable increase in temperature generated by tourists within Muierilor show cave (SE 5) was observed for the entire cold period [65] with cave lights representing a minor component. A large part of the *R. ferrumequinum* colony hibernated in the touristic sector. It may suggest that the animals preferred higher temperatures or were constrained to choose those areas by other unaccounted restrictions [89]. Human disturbances were strong, with halogen lighting systems projected on the hibernating colonies, noise, and flash photography. During the end of the hibernation, the colonies moved further away from the touristic sector to colder areas, most likely to conserve energy. Correlations between tourist passes and bat abundances per season were mainly not significant, suggesting that the touristic activity may be partially tolerated, as Dragu also mentioned [90]. One positive correlation was found for *R. hipposideros* during the beginning of hibernation; the species preferred higher torpor temperatures and maintained close proximity to the entrances, where tourists accessed the cave. This was also found by Zukal et al. who mentioned that the animals were not negatively impacted by tourism [78]. The current SDM models predicted these preferences, concentrating favourable patches in the touristic sector. Although temporary observations show no significant impact, these microclimatic preference changes can lead to an increase of artificial arousals and can on overall decrease bat body fitness on a long-term basis [79]. Specific microclimatic reconstruction projects need to be undertaken in some sectors to reduce the animal's dependency on habitat patches affected by the anthropogenic temperature changes [14]. A third small entrance in this cave that leads to a chamber that was used by hibernating bats (the Altar Hall), as proved by the guano deposits and prior observations, was closed in the past. The entrance was recently opened and then filled with rubble, but the air circulation most likely continues to flow, as ice formations can be seen near the entrance of this chamber. Sealing this opening might increase air temperature in the Altar Hall and restore a suitable area for hibernation. Bats hibernating in the touristic area might return to the previously favourable sector, decreasing the existing impact.

Valea Cetății show cave (SE 4) is used as a hibernaculum and harbours small bat populations. It was repurposed for tourism in 2010, and no significant population changes have been observed compared to the initial state [91]. Here, we recommend limiting cultural activities, such as concerts in the cave, during the hibernation period.

Despite the fact that Meziad show cave (SE 10) has a much larger volume than the previously described sites and it is fitted with an adequate low light system pointing downward, with no significant anthropogenic temperature spikes recorded during the monitoring period, the maternity colony from the upper level shows large population fluctuations which the touristic activity may cause. Lights in the upper level, close to the colony, should be further dimmed and managers should bypass the touristic flux in this area as much as possible to reduce stress.

The non-touristic caves located in remote regions (Stogu and Lacul Verde—SEs 7 and 8) did not show any touristic risk, but the more accessible sites, such as Gura Dobrogei cave, Canaraua Fetii mine and Hagieni mine (SEs 1, 2, and 3) were subject to vandalism and need to be closed off by special gating projects, because they concentrate large bat populations, which are crucial for the steppe bioregion. The Canaraua Fetii (SE 2) and Hagieni tunnels (SE 3) harbour large colonies and are an example of critically important bat artificial roosts. The Hagieni tunnel is mainly used for nurseries, as the SDM models and field observations suggest [92].

The Cloşani cave (SE 9) is a gated research site with controlled access. Therefore, the *R. euryale* hibernation colony is not submitted to significant impact, although their numbers have greatly fluctuated during recent years.

Identifying temporal and spatial bat dynamics in subteranean environments via species distribution models can help minimise accidental arousals and identify areas where ecological reconstruction techniques are needed to restore the microclimatic conditions in specific roosts [15]. As stable microclimates in subteranean environemts are changing in response to exterior temperatures [21], potentially restricting suitable habitats for cave-dwelling species [48], there is an urgent need for habitat suitability evaluations which can aid future conservation efforts. Using the existing climate change scenarios (IPCC) and the fact that temperatures fluctuations are one of the most important limiting factors for most cave-dwellers, future research can create site specific scenarios of microclimatic changes to study the distribution and occupancy of these animals [85]. Our approach can be used as a framework where species distribution modeling advancements can help decision makers apply conservation measures in light of the growing anthropogenic pressures and climate change effects, enhancing management practices for subteranean environments.

## Supporting information

**S1 File.**
(DOCX)

## Acknowledgments

During the field work we had help in securing and collecting relevant information from Răzvan Arghir, Alexandru Petculescu, Marius Robu, Ruxandra Năstase-Bucur, Daniela Cociuba, Virgil Drăguşin, and Marius Kenesz to whom we are very grateful. For access to the speleological archives and data we would like to thank Marius Vlaicu. We would like to also thank Stelian Grigore, Cristinel Fofirică, Arhur Dăscălescu, and Marius Iliescu (Hades Speleological Club, Romania) for providing the Muierilor cave map. We would also like to thank the cave administrators who have offered access and other logistical support in the show caves and the team of the SEOPMM Oceanic Club NGO for logistical support at the south-eastern sites. We thank Laurenţiu Rozylowicz and the Geographical Student Association (ASG—University of Bucharest) for help in the initial stages of the project. We would like to thank the editors and reviewers for their efforts, acknowledging that their suggestions greatly enhanced this manuscript.

## Author Contributions

**Conceptualization:** Dragoş Ştefan Măntoiu, Ionuţ Cornel Mirea, Oana Teodora Moldovan.

**Data curation:** Dragoş Ştefan Măntoiu.

**Formal analysis:** Dragoş Ştefan Măntoiu, Ionuţ Cornel Mirea, Alina Georgiana Cîşlariu, Oana Teodora Moldovan.

**Funding acquisition:** Dragoş Ştefan Măntoiu, Silviu Constantin, Oana Teodora Moldovan.

**Investigation:** Dragoş Ştefan Măntoiu, Ionuţ Cornel Mirea, Silviu Constantin, Oana Teodora Moldovan.

**Methodology:** Dragoş Ştefan Măntoiu, Ionuţ Cornel Mirea, Ionuţ Cosmin Şandric, Silviu Constantin, Oana Teodora Moldovan.

**Project administration:** Dragoş Ştefan Măntoiu, Silviu Constantin, Oana Teodora Moldovan.

**Resources:** Dragoş Ştefan Măntoiu, Ionuţ Cornel Mirea, Ionuţ Cosmin Şandric, Oana Teodora Moldovan.

**Software:** Dragoş Ştefan Măntoiu, Ionuţ Cornel Mirea, Ionuţ Cosmin Şandric, Alina Georgiana Cîşlariu, Iulian Gherghel.

**Supervision:** Dragoş Ştefan Măntoiu, Ionuţ Cosmin Şandric, Alina Georgiana Cîşlariu, Oana Teodora Moldovan.

**Validation:** Dragoş Ştefan Măntoiu, Ionuţ Cornel Mirea, Ionuţ Cosmin Şandric, Alina Georgiana Cîşlariu, Iulian Gherghel, Oana Teodora Moldovan.

**Visualization:** Dragoş Ştefan Măntoiu, Ionuţ Cosmin Şandric, Iulian Gherghel.

**Writing – original draft:** Dragoş Ştefan Măntoiu, Ionuţ Cornel Mirea, Ionuţ Cosmin Şandric, Alina Georgiana Cîşlariu, Oana Teodora Moldovan.

**Writing – review & editing:** Dragoş Ştefan Măntoiu, Alina Georgiana Cîşlariu, Silviu Constantin, Oana Teodora Moldovan.

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
