## [Decision Letter · Decision Letter 0]

16 Jun 2022

PONE-D-22-13499Bat dynamics modelling as a tool for conservation management in show cavesPLOS ONE

Dear Dr. Măntoiu,

Thank you for submitting your manuscript to PLOS ONE. After careful consideration, we feel that it has merit but does not fully meet PLOS ONE’s publication criteria as it currently stands. Therefore, we invite you to submit a revised version of the manuscript that addresses the points raised during the review process.

Both reviewers agree that this manuscript presents an interesting novel technique but also express some criticism that the authors should address in their revision. This includes the addition of background on the the conservation status of caves and bat caves in the introduction, earlier introduction of the relevant bat species but also restructuring of the discussion. The latter should include a more thorough evaluation of the potential for the application of the novel methods for other sites both locally and globally as well as the transferability of conclusions drawn based on the particular bat species that were the focus of the current study. Please refer  reviewer feedback below for more detailed comments. 

We look forward to receiving your revised manuscript.

Kind regards,

Heike Lutermann, PhD

Academic Editor

PLOS ONE

Journal Requirements:

2. We note that Figures 1-5 and S1-S3 in your submission contain [map/satellite] images which may be copyrighted. All PLOS content is published under the Creative Commons Attribution License (CC BY 4.0), which means that the manuscript, images, and Supporting Information files will be freely available online, and any third party is permitted to access, download, copy, distribute, and use these materials in any way, even commercially, with proper attribution. For these reasons, we cannot publish previously copyrighted maps or satellite images created using proprietary data, such as Google software (Google Maps, Street View, and Earth). For more information, see our copyright guidelines: http://journals.plos.org/plosone/s/licenses-and-copyright.

a) You may seek permission from the original copyright holder of Figures 1-5 and S1-S3 to publish the content specifically under the CC BY 4.0 license.  

Reviewers' comments:

Reviewer's Responses to Questions

**Comments to the Author**

1. Is the manuscript technically sound, and do the data support the conclusions?

Reviewer #1: Partly

Reviewer #2: Yes

2. Has the statistical analysis been performed appropriately and rigorously? 

Reviewer #1: Yes

Reviewer #2: Yes

3. Have the authors made all data underlying the findings in their manuscript fully available?

Reviewer #1: Yes

Reviewer #2: Yes

4. Is the manuscript presented in an intelligible fashion and written in standard English?

Reviewer #1: Yes

Reviewer #2: Yes

5. Review Comments to the Author

Reviewer #1: I greatly appreciate the opportunity to have read your manuscript entitled “Bat dynamics modelling as a tool for conservation management in show caves”. I quite found the research itself very exciting due to the specific recommendations for management actions, and these recommendations were derived using information acquired by ensemble SDMs. Ensemble SDMs are a technique that I have not read about, so I was happy to learn a new modelling method. I think that the research is very fascinating and is a great addition to the literature. However, I feel that there are portions of the paper that might require a bit of restructuring and editing. I have written my recommendations below. Hopefully they are useful to you.

Title: What I find interesting is that the title seems geared toward just show caves, but some of the sites you monitored were mines and other non-show caves. You might want to consider adjusting the title to make it less restrictive. The methods used can be applied for both non-tourist and tourist sites, and that’s neat!

Abstract: I think that the abstract clearly stated what the research was about, had clear justification and a nice summary of results. Very succinct. I have a couple minor edits:

Line 48: I would say (e.g., caves and mines) as bats are in other locations too. I do know that your paper is just about caves and mines, but I think it’s fine for the abstract to add in the “e.g.”

Line 50: You can remove “may” and just say “if optimal patches become unavailable”

Introduction: The introduction was well-written. I think it provided good background information to introduce the topic. However, I think perhaps you could briefly mention a little bit about ensemble SDMs. I noted that it was said why there are pitfalls to SDM modelling of SEs, but then ensemble SDMs are just mentioned as the method. Perhaps you could split up line 143 and 144, so that in the second sentence you can say something very brief about it (e.g., the benefit of this model type). The Woodman et al. 2019 manuscript that describes the ensemble SDM could be good here perhaps.

Here, and throughout, check that you use the hypen for “cave-dwelling” and “crevice-dwelling” to stay consistent.

Line 75: should it be “and” instead of “or” when you mention RELCOM? E.G: “and RELCOM 2007”

Line 76: “cave-dwelling bat species” – remove the ‘s’ on bats

Line 86: just nitpicky, but “select” could be used instead of “choose”

Line 118: missing a space between “et al. 1998”

Line 142: remove a period

Line 149: Something here is amiss. Perhaps it should say “This novel approach was challenged by a series of limiting factors…”

Methods: I keep wondering if already in the methods the species should be mentioned as the analyses and results are species-specific. I think this is my largest suggestion for the methods section. I only mention this (and probably state it below a few times) as I was made aware of these species in the results and it was unexpected.

Check the use of hyphens versus en-dash throughout. The en-dash is used for ranges (e.g. September–October; e.g. line 166).

I might recommend putting more information into the table descriptions so that they better “stand alone” unless the journal says otherwise.

Line 156: “We chose” instead of “have chosen”

Line 160/161: I would say in one sentence that you used flash photography. I would then state in another sentence that the work was permitted by the Romanian Academy. Two different topics.

Line 162: I think this should come before the sentence on permitting.

Line 169: I think that the years sampled could potentially fit into Table 1. Then you can just say “…and each SE was observed for two consecutive years between 2010 and 2016 (Table 1)”

Line 170: What do you mean by “later period”?

Line 171: I am not sure what you mean when you say “It was considered relevant”. Or perhaps, what is the purpose of this sentence? Are you trying to say that surveys were useful for model development?

Line 173: I might say “The spatial extend of SEs was…”

Line 174: Suggested edit: “The spatial extent of SE 1–3 was performed…”

Line 178: What is the purpose of merging the maps in a single layout if the maps have different spatial references?

Line 179: “mapping bat occurrences”? Might add “bat” for clarity

Line 187: I would rewrite and say “We used the number of passes recorded by people counters (tourist passes) as a proxy for the number of tourists who visited the caves.

Line 189: change “since” to “because”. Since refers to time

Line 190: you say you considered attributing 100 passes, but you did do this. I would say “We attributed 100 human passes per year for scalability”

Line 199: the degree symbol is before the C. should be flipped. Also the C is missing from the other value

Line 218: change “holds” to “held”

Line 222: Suggested edit “We included data on human-induced temperature changes (resulting from tourist presence and light systems) collected from show caves used in our study in our approach…”. Then you can get rid of the sentence on line 224.

Line 224–233: All the information about temperatures read to me as results. I recommend moving.

Line 244: I use a citation for the > 10 for VIF (e.g., Bowerman, B. L., and R. T. O’Connell. 1990. Linear statistical models: an applied approach. 2nd ed. PWS Kent Publishers, Belmont, California, Duxbury.)

Line 265: “coefficient, SSDM R package” just remove the hyphen and use a comma

Line 268: remove “have”, e.g., “we used a ten-percentile..”

Line 273: This is a result, and I would reference table 2 with this.

Line 273/274: This makes me wonder if the species should already be mentioned in the methods given how the models were developed.

Line 275: Move to results section. I would also then take the last two sentences of that paragraph and put into the previous paragraph (perhaps just add to the end of the previous paragraph).

Results: I think this section needs a bit of work so that it’s a bit easier to follow. I believe there are some methods mixed into the Results section that hinder the clarity. I’ve tried to identify where this happens. I also found that perhaps some of the analyses were not stated earlier, or at least were not clear as I could not match them to the results. Check for consistency on how you write the “p” for “P-value” as sometimes it’s capitalized and other times its lower case and italicized.

Line 282: This sentence is a method if you are saying that they were considered a single group for modelling purposes. If that is the case, then the species might need to be identified in the methods section.

Line 286: change “has been” to “was”

Line 288: I think the part about selecting high temperature range variation is fine, but mentioning “to raise young” seems like a point for discussion and not really a result. I might just remove this part.

Line 294–297: This reads as methods to me.

Line 297: What do you mean by “acceptable values”? I would elaborate on this sentence in general.

Line 303: This again reads as a method. The next sentence is the result of the method.

Line 306: The friedman’s test was only mentioned for the number of species in each SE. Might need to check the statistical analysis section if you did an analysis on number of bats. Also, was this summed across all SEs or for individual SEs? I would clarify that in the analysis section too.

Line 320: This is a method.

Line 332/335: I would just list the species rather than saying “most”. Or say, “all species except XX”

Line 358: So, I think I need the justification as to why the models were summarized this way. Perhaps this relates to the statements in the discussion (see line 429)?

Table 3, 4: Check how many decimal places you use for the P-values. I might recommend also matching to the results section to be consistent.

Figures 3–5: I think it would be beneficial to add to the description what the values mean, especially for someone who is unfamiliar with this type of modelling.

Discussion: This section I had the most difficulties with. The analyses, as I followed, were very species-specific, and yet I am lacking a discussion on these species-specific results. Even a discussion on the clustering of species within the sites could be better discussed. I am going to assume that the recommendations for the management of each SE are derived in part from these species, and so I would recommend discussing what you found for the bat species. Further, there are some results in the discussion that should be moved. I would also recommend not using the acronyms/letters in the discussion. I would just say “early hibernation” for instance. It makes it a bit easier to read. I think those are fine to use in the methods and results.

Line 384 and 386: I think references are needed.

Paragraph starting on line 387: This all reads as results to me. You are mentioning your findings. In the discussion you should say how these findings broadly relate to the literature. You have species-specific info here: do your results match other studies? I think the last two sentences are discussion points.

Line 401: I would expand. Break the sentence apart and reference some literature on the benefits of warm areas for bats during maternity. Could compare to how low temperatures are perhaps not ideal for rearing pups.

Line 405, and most of this paragraph, read as methods. It’s the justification for why you ran the models you did. This section is listed as “Species distribution models” and yet there is no discussion on the outcomes of the models themselves in relation to the current literature.

Is the paragraph starting with line 422 about the pros/cons to SDM versus the ensemble modelling?

Line 429–432: Methods and results. Please move to appropriate areas.

Paragraph starting with line 433: I think there are good things here, but that some more relation to the literature is needed. For example, you mention that the minimum temperature is important at the beginning of hibernation. Is this result the same seen in the literature?

Line 441: Is the statement about optimal torpor temps your finding or something from literature? If literature, it needs a citation.

Paragraph starting line 447 is results. Discuss more, for example, why you think the SDM values are higher in more complex systems.

Line 452: What do you mean by “sensible species”?

Line 453–455: methods; also I think you mean “susceptible to human disturbance” and not “sensible” here

Line 470: this seems to be a description of a site (method) rather than a discussion point

Lines 473–493: I am just not really seeing a discussion with the literature. Some of this is a description of the site which would go in the methods.

Reviewer #2: I enjoyed reading the manuscript and I believe this is an important innovations in bat cave monitoring using a cost-effective approach. The applications of SDM to cave and subterranean prioritization is gaining such attention because of its huge potential to project potential areas for conservation and protection.

Măntoiu et al. presented a detailed approaches and analysis of field data. It provides a useful guideline and techniques that benefits many bat cave conservation biologists. However, there are key things I wish to suggest that I believe will improve the narrative and impact of their work. First, in the introduction there is a lack of background on the state of conservation status of caves and bat caves, which is very important to really convey the importance of the work and it's implication. You may consider these new references

How bats are dependent to caves?

https://www.nature.com/articles/s41597-022-01234-4

Tanalgo, K. C., Tabora, J. A. G., de Oliveira, H. F. M., Haelewaters, D., Beranek, C. T., Otálora-Ardila, A., ... & Hughes, A. C. (2022). DarkCideS 1.0, a global database for bats in karsts and caves. Scientific Data, 9(1), 1-12.

State of conservation of caves

https://onlinelibrary.wiley.com/doi/full/10.1111/brv.12851

Mammola, S., Meierhofer, M. B., Borges, P. A., Colado, R., Culver, D. C., Deharveng, L., ... & Cardoso, P. (2022). Towards evidence‐based conservation of subterranean ecosystems. Biological Reviews.

https://www.science.org/doi/10.1126/science.abo1973

Ferreira, R. L., Bernard, E., da Cruz Júnior, F. W., Piló, L. B., Calux, A., Souza-Silva, M., ... & Frick, W. F. (2022). Brazilian cave heritage under siege. Science, 375(6586), 1238-1239.

Also, mention the ecosystem services of cave bats and what is their current status in Romania.

It is also wise to discuss what are current techniques and approaches applied to prioritize bat caves for conservation, this idea is somewhat lost in the current of the paper. It's good to inform your readers what are other approaches and how your current approach is unique and how it can be integrated to the ones that are currently existing. What are the limitations of both previous and current prioritization? What are the elements they have considered?

The new approach was applied in the temperate region, is it possible to be applied in the tropics too where bat cave dynamics is a bit different? It's good to discuss this limitation and potential too.

The paper started with a good and exciting narrative but the closing narrative needs a stronger conclusion and it's also worth mentioning the important caveats of the present work.

I hope my suggestions are useful and I look forward for the revised version of this paper.

6. PLOS authors have the option to publish the peer review history of their article (what does this mean?). If published, this will include your full peer review and any attached files.

Reviewer #1: No

Reviewer #2: No

---

## [Author Response · Author response to Decision Letter 0]

25 Aug 2022

Reviewer #1: I greatly appreciate the opportunity to have read your manuscript entitled “Bat dynamics modelling as a tool for conservation management in show caves”. I quite found the research itself very exciting due to the specific recommendations for management actions, and these recommendations were derived using information acquired by ensemble SDMs. Ensemble SDMs are a technique that I have not read about, so I was happy to learn a new modelling method. I think that the research is very fascinating and is a great addition to the literature. However, I feel that there are portions of the paper that might require a bit of restructuring and editing. I have written my recommendations below. Hopefully they are useful to you.

Authors: Dear Reviewer, We would like to thank you very much for your time in reviewing our paper and for your valuable recommendations. We are very glad that you acknowledged our research and for considering it would be a great addition to the literature.

We have changed large sections of our manuscript according to your suggestions and provided below, point by point, the edits we made. We attached to these comments two versions of our revised manuscript, one version with track changes and the other one with all changes accepted. Within the comments, we mentioned the lines where you can find the modifications in both versions of the revised manuscript.

Reviewer: Title: What I find interesting is that the title seems geared toward just show caves, but some of the sites you monitored were mines and other non-show caves. You might want to consider adjusting the title to make it less restrictive. The methods used can be applied for both non-tourist and tourist sites, and that’s neat!

Authors: We adjusted the title, generally referring further to the caves (both show- and non-show caves) and mines as subterranean environments.

Reviewer: Abstract: I think that the abstract clearly stated what the research was about, had clear justification and a nice summary of results. Very succinct. I have a couple minor edits:

Line 48: I would say (e.g., caves and mines) as bats are in other locations too. I do know that your paper is just about caves and mines, but I think it’s fine for the abstract to add in the “e.g.”

Authors: we added e.g., caves and mines, as you suggested (line 49 - version with track changes / line 37 – version with all changes accepted).

Reviewer: Line 50: You can remove “may” and just say “if optimal patches become unavailable”

Authors: We removed the word may (line 51 – version with track changes / line 39 – version with all changes accepted).

Reviewer: Introduction: The introduction was well-written. I think it provided good background information to introduce the topic. However, I think perhaps you could briefly mention a little bit about ensemble SDMs. I noted that it was said why there are pitfalls to SDM modelling of SEs, but then ensemble SDMs are just mentioned as the method. Perhaps you could split up line 143 and 144, so that in the second sentence you can say something very brief about it (e.g., the benefit of this model type). The Woodman et al. 2019 manuscript that describes the ensemble SDM could be good here perhaps.

Authors: We added in the Introduction information regarding the advantage of applying ensemble species distribution modeling (ESDM) (lines 172-175 – version with track changes / lines 142-145 – version with all changes accepted). Also, as the second Reviewer suggested, we added some new paragraphs in the Introduction section regarding the conservation status of caves and cave-dwelling bats, as well as the ecosystem services they provide, stating therefore, the importance of establishing specific conservation measures to ensure the long-term survival of cave-dwelling bats.

Reviewer: Here, and throughout, check that you use the hypen for “cave-dwelling” and “crevice-dwelling” to stay consistent.

Authors: We checked throughout the text and used the hypen for cave-dwelling / crevice-dwelling.

Reviewer: Line 75: should it be “and” instead of “or” when you mention RELCOM? E.G: “and RELCOM 2007”

Authors: We changes or with and (line 83 – version with track changes / line 67 – version with all changes accepted).

Reviewer: Line 76: “cave-dwelling bat species” – remove the ‘s’ on bats

Authors: We removed the ‘s’ from bats (line 88 – version with track changes / line 72 – version with all changes accepted).

Reviewer: Line 86: just nitpicky, but “select” could be used instead of “choose”

Authors: We rephrased this sentence (lines 102-105 – version with track changes / lines 85-87 – version with all changes accepted).

Reviewer: Line 118: missing a space between “et al. 1998”

Authors: We modified the citation style according to the Journal requirements, which solved the issue you addressed.

Reviewer: Line 142: remove a period

Authors: We removed the period (line 158 – version with track changes / line 129 – version with all changes accepted).

Reviewer: Line 149: Something here is amiss. Perhaps it should say “This novel approach was challenged by a series of limiting factors…”

Authors: We corrected the sentence (line 180 – version with track changes / line 149 – version with all changes accepted).

Reviewer: Methods: I keep wondering if already in the methods the species should be mentioned as the analyses and results are species-specific. I think this is my largest suggestion for the methods section. I only mention this (and probably state it below a few times) as I was made aware of these species in the results and it was unexpected.

Authors: We added a paragraph in the Methods section where we listed the focal bat species for our study (lines 230-235 – version with track changes / lines 188-192 – version with all changes accepted).

Reviewer: Check the use of hyphens versus en-dash throughout. The en-dash is used for ranges (e.g. September–October; e.g. line 166).

Authors: We checked and corrected the hyphens vs. en-dash throughout the text.

Reviewer: I might recommend putting more information into the table descriptions so that they better “stand alone” unless the journal says otherwise.

Authors: We added in Table 1 another column with the sampled period per each analysed site, as suggested below. We consider that all the relevant information on the analysed sites are already included within the table.

Reviewer: Line 156: “We chose” instead of “have chosen”

Authors: We corrected the sentence (line 188 – version with track changes / line 157 – version with all changes accepted).

Reviewer: Line 160/161: I would say in one sentence that you used flash photography. I would then state in another sentence that the work was permitted by the Romanian Academy. Two different topics.

Authors: We made the change according to your suggestions (see lines 196-200 – version with track changes / lines 163-167 – version with all changes accepted). 

Reviewer: Line 162: I think this should come before the sentence on permitting.

Authors: We moved the sentence before the one on permitting (lines 198-199 – version with track changes / lines 165-166 – version with all changes accepted).

Reviewer: Line 169: I think that the years sampled could potentially fit into Table 1. Then you can just say “…and each SE was observed for two consecutive years between 2010 and 2016 (Table 1)”

Authors: We fitted the table with the analysed period per each site and changed the sentence accordingly (lines 207-208 – version with track changes / lines 172-173 – version with all changes accepted).

Reviewer: Line 170: What do you mean by “later period”?

Authors: Here we referred to the 2014-2016 period. In the revised version, we mentioned the exact period we were referring to (line 208 – version with track changes / line 173 – version with all changes accepted.

Reviewer: Line 171: I am not sure what you mean when you say “It was considered relevant”. Or perhaps, what is the purpose of this sentence? Are you trying to say that surveys were useful for model development?

Authors: We referred to the fact that although the period 2014-2016 did not cover the complete maternity interval, we considered these data relevant for our model development, as within this period, the species already have formed colonies. We rephrased the sentence (lines 209-210 – version with track changes / lines 174-175 – version with all changes accepted).

Reviewer: Line 173: I might say “The spatial extend of SEs was…”

Authors: We made the modification according to your suggestion (line 212 – version with track changes / line 176 – version with all changes accepted).

Reviewer: Line 174: Suggested edit: “The spatial extent of SE 1–3 was performed…”

Authors: We rephrased the previous sentence, so your suggested edit no longer applies here. We further mentioned that the laser meter was used for all cave surveys (lines 214-216 – version with track changes / lines 178-179 – version with all changes accepted).

Reviewer: Line 178: What is the purpose of merging the maps in a single layout if the maps have different spatial references?

Authors: We merged the maps into a single layout only as a visual aid for results representation (so we avoid including within the manuscript 10 figures), while the models were further performed at their original scale of all SEs. We also added this information within the text (lines 220-223 – version with track changes / lines 181-183 – version with all changes accepted).

Reviewer: Line 179: “mapping bat occurrences”? Might add “bat” for clarity

Authors: We added the word bat, as you suggested (line 236 – version with track changes / line 193 – version with all changes accepted).

Reviewer: Line 187: I would rewrite and say “We used the number of passes recorded by people counters (tourist passes) as a proxy for the number of tourists who visited the caves.

Authors: We made the edit accordingly (see lines 244-246 – version with track changes / lines 200-201 – version with all changes accepted).

Reviewer: Line 189: change “since” to “because”. Since refers to time

Authors: We replaced since with because (line 247 – version with track changes / line 202 – version with all changes accepted).

Reviewer: Line 190: you say you considered attributing 100 passes, but you did do this. I would say “We attributed 100 human passes per year for scalability”

Authors: We made the change according to your suggestion (line 249 - version with track changes / line 203 - version with all changes accepted).

Reviewer: Line 199: the degree symbol is before the C. should be flipped. Also the C is missing from the other value

Authors: We made the corrections (line 257 - version with track changes / lines 210-211 version with all changes accepted) .

Reviewer: Line 218: change “holds” to “held”

Authors: We changed the term (line 278 - version with track changes / line 231 - version with all changes accepted)

Reviewer: Line 222: Suggested edit “We included data on human-induced temperature changes (resulting from tourist presence and light systems) collected from show caves used in our study in our approach…”. Then you can get rid of the sentence on line 224.

Authors: We made the edit accordingly (see lines 288-289 - version with track changes / lines 238-239 - version with all changes accepted).

Reviewer: Line 224–233: All the information about temperatures read to me as results. I recommend moving.

Authors: We moved these to the results section (lines 405-413 - version with track changes / lines 314-322 - version with all changes accepted).

Reviewer: I use a citation for the > 10 for VIF (e.g., Bowerman, B. L., and R. T. O’Connell. 1990. Linear statistical models: an applied approach. 2nd ed. PWS Kent Publishers, Belmont, California, Duxbury.)

Authors: We cited the reference you suggested (line 317 - version with track changes / line 255 - version with all changes accepted).

Reviewer: Line 265: “coefficient, SSDM R package” just remove the hyphen and use a comma

Authors: We made the edit (line 341 - version with track changes / line 276 - version with all changes accepted).

Reviewer: Line 268: remove “have”, e.g., “we used a ten-percentile..”

Authors: We removed have (line 344 - version with track changes / line 279 - version with all changes accepted).

Reviewer: Line 273: This is a result, and I would reference table 2 with this.

Authors: We added this sentence, along with the next one at the end of the previous paragraph (lines 329-331 - version with track changes / lines 266-268 - version with all changes accepted), as you suggested below. We referenced table 2 to the first sentence. The sentence is relevant here because it shows that the models were performed for all of the sites, regalrdless of species presence on site. 

Reviewer: Line 273/274: This makes me wonder if the species should already be mentioned in the methods given how the models were developed.

Authors: As we mentioned above, we listed in the revised version of the manuscript the analysed bat species within the methods section. 

Reviewer: Line 275: Move to results section. I would also then take the last two sentences of that paragraph and put into the previous paragraph (perhaps just add to the end of the previous paragraph).

Authors: We moved these last two sentences at the end of the previous paragraph (lines 330-332 - version with track changes / lines 266-268 - version with all changes accepted) and we moved the mentioned line at the beginning of the results chapter (lines 460-461 - version with track changes / lines 366-367 - version with all changes accepted)

Reviewer: Results: I think this section needs a bit of work so that it’s a bit easier to follow. I believe there are some methods mixed into the Results section that hinder the clarity. I’ve tried to identify where this happens. I also found that perhaps some of the analyses were not stated earlier, or at least were not clear as I could not match them to the results.

Authors: While reading the suggestions you provided, we realised that we indeed have mixed some methods with the results, therefore we thank you for the edits you proposed. We followed all of them and hopefully now, this section is easier to follow.

Reviewer: Check for consistency on how you write the “p” for “P-value” as sometimes it’s capitalized and other times its lower case and italicized.

Authors: We capitalised the P throughout the text.

Reviewer: Line 282: This sentence is a method if you are saying that they were considered a single group for modelling purposes. If that is the case, then the species might need to be identified in the methods section.

Authors: We moved this entire paragraph to the methods section such as to respond to your above suggestion regarding also mentioning the analysed species in the data collection section (see lines 230-235 - version with track changes / lines 188-192 - version with all changes accepted).

Reviewer: Line 286: change “has been” to “was”

Authors: We replaced has been with was (line 359- version with track changes / line 289 - version with all changes accepted).

Reviewer: Line 288: I think the part about selecting high temperature range variation is fine, but mentioning “to raise young” seems like a point for discussion and not really a result. I might just remove this part.

Authors: We rephrased the sentence and removed the part on raising the young in the revised version of our manuscript.

Reviewer: Line 294–297: This reads as methods to me.

Authors: We rephrased the paragraph and moved it to the methods section (lines 279-286 - version with track changes / lines 232-237 - version with all changes accepted).

Reviewer: Line 297: What do you mean by “acceptable values”? I would elaborate on this sentence in general.

Authors: We changed large sections of this sub-chapter in the revised version of our manuscript, please see lines 371 – 413 - version with track changes / lines 298 – 322 - version with all changes accepted. 

Reviewer: Line 303: This again reads as a method. The next sentence is the result of the method.

Authors: We moved this sentence to the methods section (lines 279-286 - version with track changes / lines 232-237 - version with all changes accepted) and rephrased large areas of the Air temperature monitoring and environmental datasets sub-chapter from the Results.

Reviewer: Line 306: The friedman’s test was only mentioned for the number of species in each SE. Might need to check the statistical analysis section if you did an analysis on number of bats. Also, was this summed across all SEs or for individual SEs? I would clarify that in the analysis section too.

Authors: We elaborated on the statistical analysis in the Methods section in the revised version of our manuscript (please see lines 302-306 - version with track changes / lines 241-244 - version with all changes accepted).

Reviewer: Line 320: This is a method.

Authors: We removed this sentence from the Results section.

Reviewer: Line 332/335: I would just list the species rather than saying “most”. Or say, “all species except XX”

Authors: We made the edit accordingly (see lines 435-443 - version with track changes / lines 343-349 - version with all changes accepted).

Reviewer: Line 358: So, I think I need the justification as to why the models were summarized this way. Perhaps this relates to the statements in the discussion (see line 429)?

Authors: We summed the binary models per period such as to identify the areas where an aggregation of species was more likely to appear. This approach does not relate to the statement you addressed in the discussion. That mention was made only from a methodological aproach.

Reviewer: Table 3, 4: Check how many decimal places you use for the P-values. I might recommend also matching to the results section to be consistent.

Authors: We made the edit such as to have the same number of decimals.

Reviewer: Figures 3–5: I think it would be beneficial to add to the description what the values mean, especially for someone who is unfamiliar with this type of modelling.

Authors: we have added the following mention to each figure capture: Sum of SDMs shows the number of suitable habitats which overlap within a SE.

Reviewer: Discussion: This section I had the most difficulties with. The analyses, as I followed, were very species-specific, and yet I am lacking a discussion on these species-specific results. Even a discussion on the clustering of species within the sites could be better discussed. I am going to assume that the recommendations for the management of each SE are derived in part from these species, and so I would recommend discussing what you found for the bat species. Further, there are some results in the discussion that should be moved. I would also recommend not using the acronyms/letters in the discussion. I would just say “early hibernation” for instance. It makes it a bit easier to read. I think those are fine to use in the methods and results.

Authors: We have revised large sections of this chapter according to your suggestions, so we hope you will now find our approach more appropriate. We also added a new sub-chapter Ecology of the focal bat species, where we address our species-specific results to those from the literature.

Reviewer: Line 384 and 386: I think references are needed.

Authors: We added the needed references (lines 543 - version with track changes / lines 440 - version with all changes accepted and line 545 - version with track changes / line 442 - version with all changes accepted).

Reviewer: Paragraph starting on line 387: This all reads as results to me. You are mentioning your findings. In the discussion you should say how these findings broadly relate to the literature. You have species-specific info here: do your results match other studies? I think the last two sentences are discussion points.

Authors: We removed this paragraph (except for the last two phrases) from the discussion section and integrated the results within the appropriate sections (lines 450-458 - version with track changes / lines 356-364 version with all changes accepted). We further related our findings to those from the literature within the Ecology of the focal bat species section, found in the Discussions chapter.

Reviewer: Line 401: I would expand. Break the sentence apart and reference some literature on the benefits of warm areas for bats during maternity. Could compare to how low temperatures are perhaps not ideal for rearing pups.

Authors: We elaborated on this idea on lines 544-551 - version with track changes / lines 441-448 - version with all changes accepted.

Reviewer: Line 405, and most of this paragraph, read as methods. It’s the justification for why you ran the models you did. This section is listed as “Species distribution models” and yet there is no discussion on the outcomes of the models themselves in relation to the current literature.

Authors: We have re-written the section on Species distribution models, discussing the outcome of our models in relation to the current literature (starting with line 576 - version with track changes / line 454 - version with all changes accepted). Also, the sections of the paragraph you are referring to, that reads as methods has been moved to the appropriate area from the Methods (starting with line 269 - version with track changes / line 222 - version with all changes accepted).

Reviewer: Is the paragraph starting with line 422 about the pros/cons to SDM versus the ensemble modelling?

Authors: This paragraph refer to the fact that when this modeling technique is applied on a smaller prediction area, characterised by fewer fluctuations of the environmental conditions, it generates more accurate results than when it is applied at a regional or continental scale, as it provides a better control on the sampling and validation strategies. We rephrased the paragraph and we hope now it states clearer what we were referring to. Please see lines 603-622 - version with track changes / lines 461-474 - version with all changes accepted.

Reviewer: Line 429–432: Methods and results. Please move to appropriate areas.

Authors: We moved the paragraph to lines 328-330 - version with track changes / lines 264-266 - version with all changes accepted.

Reviewer: Paragraph starting with line 433: I think there are good things here, but that some more relation to the literature is needed. For example, you mention that the minimum temperature is important at the beginning of hibernation. Is this result the same seen in the literature?

Authors: We elaborated on this subject. Please see paragraph staring with line 634 - version with track changes / line 480 - version with all changes accepted.

Reviewer: Line 441: Is the statement about optimal torpor temps your finding or something from literature? If literature, it needs a citation.

Authors: We added a citation at line 506 version with track changes /line 408 - version with all changes accepted.

Reviewer: Paragraph starting line 447 is results. Discuss more, for example, why you think the SDM values are higher in more complex systems.

Authors: We removed the paragraph from the discussion section.

Reviewer: Line 452: What do you mean by “sensible species”?

Authors: We removed the paragraph you addressed in the revised version of the manuscript.

Reviewer: Line 453–455: methods; also I think you mean “susceptible to human disturbance” and not “sensible” here

Authors: We also removed this paragraph. We repharased the idea and included it in the Methods section.

Reviewer: Line 470: this seems to be a description of a site (method) rather than a discussion point. Lines 473–493: I am just not really seeing a discussion with the literature. Some of this is a description of the site which would go in the methods.

Authors: We removed this paragraph from the Discussion section. In the revised version of our manuscript, we removed the sub-chapter SDM as tools for managing human impact in subterranean environments, as it contained mainly the description of the analysed sites and incorporated the relevant information in the Implications for the conservation management of bats in show caves sub-chapter.

We hope we have reached the implementation of your suggestions and improved our discussion section. Thank you again for your suggestions. We believe they contributed a great deal to enhancing our manuscript. We hope you will find the revised version of our manuscript as suitable for publication.

Reviewer #2: I enjoyed reading the manuscript and I believe this is an important innovations in bat cave monitoring using a cost-effective approach. The applications of SDM to cave and subterranean prioritization is gaining such attention because of its huge potential to project potential areas for conservation and protection.

Măntoiu et al. presented a detailed approaches and analysis of field data. It provides a useful guideline and techniques that benefits many bat cave conservation biologists. However, there are key things I wish to suggest that I believe will improve the narrative and impact of their work. 

Authors: Dear Reviewer, We would like to thank you very much for your time in reviewing our paper and for your valuable recommendations. We are very glad that you acknowledged our research and for considering it provides a useful guideline and techniques that benefits bat cave conservation biologists.

We have changed large sections of our manuscript according to your and the second reviewers’ suggestions and provided below, the edits we made. We attached to these comments two versions of our revised manuscript, one version with track changes and the other one with all changes accepted. Within the comments, we mentioned the lines where you can find the modifications in both versions of the revised manuscript .

Reviewer: First, in the introduction there is a lack of background on the state of conservation status of caves and bat caves, which is very important to really convey the importance of the work and it's implication. You may consider these new references

How bats are dependent to caves?

https://www.nature.com/articles/s41597-022-01234-4

Tanalgo, K. C., Tabora, J. A. G., de Oliveira, H. F. M., Haelewaters, D., Beranek, C. T., Otálora-Ardila, A., ... & Hughes, A. C. (2022). DarkCideS 1.0, a global database for bats in karsts and caves. Scientific Data, 9(1), 1-12.

State of conservation of caves

https://onlinelibrary.wiley.com/doi/full/10.1111/brv.12851

Mammola, S., Meierhofer, M. B., Borges, P. A., Colado, R., Culver, D. C., Deharveng, L., ... & Cardoso, P. (2022). Towards evidence‐based conservation of subterranean ecosystems. Biological Reviews.

https://www.science.org/doi/10.1126/science.abo1973

Ferreira, R. L., Bernard, E., da Cruz Júnior, F. W., Piló, L. B., Calux, A., Souza-Silva, M., ... & Frick, W. F. (2022). Brazilian cave heritage under siege. Science, 375(6586), 1238-1239.

Reviewer: Also, mention the ecosystem services of cave bats and what is their current status in Romania.

Authors: According to your suggestions, we added new paragraphs addressing the conservation status of caves and cave-dwelling bats, as well as the ecosystems services provided by the cave bats, within the Introduction section. In the paragraph from lines 72-80 - version with track changes / lines 59-64 - version with all changes accepted, we state the importance of subterranean environments as habitats harbouring many endemic species and their conservation status system. On lines 96-101 - version with track changes / lines 78-83 - version with all changes accepted, we provide information on the valuable ecosystem services provided by the cave-dwelling bats, elaborating on the necessity of establishing specific conservation measures to ensure their long-term survival. We mentioned the limitations of studies on cave-dwelling bats from the temperature region (lines 159-163 - version with track changes / lines 130-134 - version with all changes accepted), as well as the protection status of the analysed cave-dwelling bats, within the Introduction section (lines 164-171 - version with track changes / lines 135-141 - version with all changes accepted). We hope that, in the revised version of our manuscript, we better conveyed the importance of our study, as well as its implications for conservation.

Reviewer: It is also wise to discuss what are current techniques and approaches applied to prioritize bat caves for conservation, this idea is somewhat lost in the current of the paper. It's good to inform your readers what are other approaches and how your current approach is unique and how it can be integrated to the ones that are currently existing. What are the limitations of both previous and current prioritization? What are the elements they have considered? The new approach was applied in the temperate region, is it possible to be applied in the tropics too where bat cave dynamics is a bit different? It's good to discuss this limitation and potential too.

Authors: We have include mentions within the Introduction (lines 83-86 - version with track changes / lines 67-70 - version with all changes accepted) about some cave prioritisation, as EUROBATS curently has a standard for defining important bat roosts within the member parties, but also we discussed about the broad definitions of cave conservation statuses worldwide (lines 74-77 - version with track changes / lines 59-62 - version with all changes accepted). We later discussed about the limitations of this study and the future applications (lines 636-640 - version with track changes / lines 480-484 - version with all changes accepted). 

Reviewer: The paper started with a good and exciting narrative but the closing narrative needs a stronger conclusion and it's also worth mentioning the important caveats of the present work.

Reviewer: I hope my suggestions are useful and I look forward for the revised version of this paper.

Authors: We have revised large sections of the Discussion chapter according to both yours and the second reviewer’s suggestions, so we hope you will now find our approach more appropriate.

We tried to integrate your suggestions within different sections of the Discussion. Please read the revised version of this chapter. Also, we have re-written the sub-chapter on Species distribution models from the Discussion, where we included the limitations of our study, as you suggested. Please see lines 575-652 - version with track changes / lines 453-495 - version with all changes accepted.

We hope we have reached the implementation of your suggestions and improved our discussion section. Thank you again for your suggestions. We believe they contributed a great deal to enhancing our manuscript. We hope you will find the revised version of our manuscript as suitable for publication.

---

## [Decision Letter · Decision Letter 1]

19 Sep 2022

PONE-D-22-13499R1Bat dynamics modelling as a tool for conservation management in subterranean environmentsPLOS ONE

Dear Dr. Măntoiu,

Thank you for submitting your manuscript to PLOS ONE. After careful consideration, we feel that it has merit but does not fully meet PLOS ONE’s publication criteria as it currently stands. Therefore, we invite you to submit a revised version of the manuscript that addresses the points raised during the review process.

 The reviewers are largely satisfied but still have a few issues the authors should address in their revision. Please see feedback provided below for details.

We look forward to receiving your revised manuscript.

Kind regards,

Heike Lutermann, PhD

Academic Editor

PLOS ONE

Journal Requirements:

Reviewers' comments:

Reviewer's Responses to Questions

**Comments to the Author**

1. If the authors have adequately addressed your comments raised in a previous round of review and you feel that this manuscript is now acceptable for publication, you may indicate that here to bypass the “Comments to the Author” section, enter your conflict of interest statement in the “Confidential to Editor” section, and submit your "Accept" recommendation.

Reviewer #1: All comments have been addressed

Reviewer #2: All comments have been addressed

2. Is the manuscript technically sound, and do the data support the conclusions?

Reviewer #1: Yes

Reviewer #2: Yes

3. Has the statistical analysis been performed appropriately and rigorously? 

Reviewer #1: Yes

Reviewer #2: Yes

4. Have the authors made all data underlying the findings in their manuscript fully available?

Reviewer #1: Yes

Reviewer #2: Yes

5. Is the manuscript presented in an intelligible fashion and written in standard English?

Reviewer #1: Yes

Reviewer #2: Yes

6. Review Comments to the Author

Reviewer #1: I am grateful for the opportunity to read the second draft of your manuscript. I think this draft is greatly enhanced, and I appreciate and acknowledge the effort put in by all authors. Overall, I found that my suggestions were considered and incorporated when appropriate, and I believe the same of the second reviewer. Again, fantastic work and a very interesting study.

I just have a few small things I caught when re-reading the manuscript. One comment that can be checked throughout is that I think it’s ok to abbreviate the genus after it is written out once in each section (e.g., line 191).

In the clean copy, check the following:

Line 47: can say “management measures are proposed” instead of "were"

Check line 345 for “only R. 18euryale” – typo.

Line 250: can abbreviate species (as comment above), but also there is a typo “Rhinolophus 12uryale”

Line 318: should it be 250.000 or 250,000?

Line 436: remove the comma after “Myotis myotis / blythii”

Line 475: can remove “in order” – not needed and will shorten the sentence. In most cases, if “in order” comes in the middle of a sentence, you can remove it and the sentence will still be correct (e.g., line 133, 140, 142, 475). Optional changes of course.

Reviewer #2: I congratulate the authors for the revised version of the manuscript. I am satisfied with the revisions made by the authors and I have no further comments.

7. PLOS authors have the option to publish the peer review history of their article (what does this mean?). If published, this will include your full peer review and any attached files.

Reviewer #1: No

Reviewer #2: No

---

## [Author Response · Author response to Decision Letter 1]

25 Sep 2022

Dear reviewers,

We have addressed all the typos and errors mentioned in this new review, with one specific mention:

Reviewer#1

Line 318: should it be 250.000 or 250,000?

Response: We eliminated the coma or point within this number to avoid any misunderstandings; it was a quarter of a million.

We would like to express our sincere gratitude for the kind words and for your intense work for this manuscript. The changes you have suggested helped us greatly improve the manuscript, with constructive and concise arguments. We feel that this was one of the most helpful reviews we processed.

We have included your contributions to our acknowledgements.

Sincerely yours,

The authors

---

## [Editor Report · Decision Letter 2]

26 Sep 2022

Bat dynamics modelling as a tool for conservation management in subterranean environments

PONE-D-22-13499R2

Dear Dr. Măntoiu,

We’re pleased to inform you that your manuscript has been judged scientifically suitable for publication and will be formally accepted for publication once it meets all outstanding technical requirements.

Kind regards,

Heike Lutermann, PhD

Academic Editor

PLOS ONE
---

## [Editor Report · Acceptance letter]

12 Oct 2022

PONE-D-22-13499R2 

Bat dynamics modelling as a tool for conservation management in subterranean environments 

Dear Dr. Măntoiu:

I'm pleased to inform you that your manuscript has been deemed suitable for publication in PLOS ONE. Congratulations! Your manuscript is now with our production department. 

Kind regards, 

on behalf of

Dr Heike Lutermann 

Academic Editor

PLOS ONE